# Mechanistic insights into the R-loop formation and cleavage in CRISPR-Cas12i1

Bo Zhang [1,2], Diyin Luo[1,2], Yu Li[1,2], Vanja Perčulija [1], Jing Chen[1], Jinying Lin[1], Yangmiao Ye[1] &
Songying Ouyang [1✉]

Cas12i is a newly identified member of the functionally diverse type V CRISPR-Cas effectors. Although Cas12i has the potential to serve as genome-editing tool, its structural and functional characteristics need to be investigated in more detail before effective application. Here we report the crystal structures of the Cas12i1 R-loop complexes before and after target DNA cleavage to elucidate the mechanisms underlying target DNA duplex unwinding, R-loop formation and cis cleavage. The structure of the R-loop complex after target DNA cleavage also provides information regarding trans cleavage. Besides, we report a crystal structure of the Cas12i1 binary complex interacting with a pseudo target oligonucleotide, which mimics target interrogation. Upon target DNA duplex binding, the Cas12i1 PAM-interacting cleft undergoes a remarkable open-to-closed adjustment. Notably, a zipper motif in the Helical-I domain facilitates unzipping of the target DNA duplex. Formation of the 19-bp crRNA-target DNA strand heteroduplex in the R-loop complexes triggers a conformational rearrangement and unleashes the DNase activity. This study provides valuable insights for developing Cas12i1 into a reliable genome-editing tool.

[1] The Key Laboratory of Innate Immune Biology of Fujian Province, Provincial University Key Laboratory of Cellular Stress Response and Metabolic Regulation, Biomedical Research Center of South China, Key Laboratory of OptoElectronic Science and Technology for Medicine of Ministry of Education, College of Life Sciences, Fujian Normal University, Fuzhou, China. [2] These authors contributed equally: Bo Zhang, Diyin Luo, Yu Li. ✉email: ouyangsy@fjnu.edu.cn

CRISPR-Cas (clustered regularly interspaced short palindromic repeats and CRISPR-associated proteins) adaptive immune systems are used by bacteria and archaea for protection against phages and foreign mobile genetic elements (MGEs)[1–3]. CRISPR stores information of invading nucleic acid fragments between two repeat regions of CRISPR array as spacer sequences. CRISPR-Cas systems complete their defense tasks through a three-step process of spacer acquisition, CRISPR RNA (crRNA) biogenesis, and target interference[4,5]. CRISPR-Cas systems exhibit great diversity and are categorized into two main classes, which are further divided into different types based on the presence of signature Cas effectors[6–8]. Class 1 systems (including type I, III, and IV systems) utilize multiple small Cas effectors and crRNA to form the interference module. In contrast, class 2 systems (including type II, V, and VI systems) employ a single Cas effector and crRNA for interference. Because of their simplicity and high efficiency, type II and V effectors such as Cas9, Cas12a (Cpf1), Cas12b (C2c1), and Cas12e (CasX) have been successfully developed into powerful tools for genome editing[9–20].

In recent years, the search for new class 2 CRISPR-Cas systems has led to the discovery of new type V systems that contain Cas12c, Cas12d, Cas12g, Cas12i, Cas14a, and Cas12j effectors. These new type V systems are functionally distinct from the previously known class 2 effectors and can thus enrich the CRISPR-Cas toolkit for genome editing[21–24]. Among them, Cas12i is characterized by a relatively smaller size (1033-1093aa) compared with Cas12a. Cas12i has substantially different efficiencies in cutting two strands of target DNA, which results in a nicking activity that could be utilized to develop a double-nicking approach for genome editing with high specificity[22,25]. It has been reported that Cas12i has distant similarity to Cas12b and both effectors recognize a 5′-TTN-3′ PAM (protospacer adjacent motif). Interestingly, Cas12i autonomously processes precursor crRNA (pre-crRNA) to form mature crRNA, but Cas12b itself does not own this capability[26].

More recently, cryo-EM and crystal structures of Cas12i in crRNA-bound and crRNA-target-bound states have been reported[27,28]. However, a number of key questions concerning the target DNA duplex unwinding, R-loop formation, target DNA duplex cleavage, key residues responsible for PAM determination in Cas12i1, and the mechanism of the DNase activation remain unanswered or require further investigation. To address these questions, we report the crystal structures of the Cas12i1 ternary complexes in the R-loop states before and after target DNA cleavage. Furthermore, we report the crystal structure of the Cas12i1 binary complex, which contains a relatively flexible PAM-interacting (PI) domain. The conformational rearrangements between the binary and ternary complexes unveil how target DNA binding triggers the DNase activity of Cas12i1. Using structural analysis and biochemical experiments, critical motifs and residues related to the aforementioned questions are identified and validated. Last, we propose a multi-step model of Cas12i1 fulfilling its function. The results of this study will provide mechanistic guidance for rational employing the CRISPR-Cas12i1 system for future applications in genome editing.

## Results

**Overall architectures of two ternary complexes of Cas12i1.** To investigate how Cas12i1 recognizes, unwinds, and cleaves two strands of target DNA, we have solved two crystal structures of Cas12i1 ternary complexes. The first ternary complex, which was formed with the catalytically inactive Cas12i1 D647A mutant, represents the R-loop state before target DNA cleavage (hereafter termed the pre-cleavage R-loop complex) and its crystal structure was determined at 2.75 Å resolution (Supplementary Table 1).

The second ternary complex uses the wild-type Cas12i1 to form the R-loop complex after target DNA cleavage (hereafter termed the post-cleavage R-loop complex) and its crystal structure was determined at 2.45 Å resolution. To obtain these crystal structures, the crystal structure of the selenomethionine (SeMet)-labeled wild-type Cas12i1 ternary complex was solved using the single-wavelength anomalous diffraction method. Next, the SeMet-labeled model was used to solve the crystal structures of two unlabeled ternary complexes using the molecular replacement method. The asymmetric unit of both crystal structures contains a single ternary complex. The structures of Cas12i1 in two ternary complexes resemble each other, and the root-mean-square deviation (RMSD) value for their backbone Cα atoms is 0.52 Å. Major differences between the two complexes exist in the states of target DNA, which will be described in depth later.

Compared with the other type V effectors with known structures, the structure of Cas12i1 in the ternary complex displays significant structural divergence. The typical bilobed architecture consisting of the recognition (REC) lobe and the nuclease (NUC) lobe can be easily distinguished, though organization and distribution of individual domains in Cas12i1 are distinct from those of Cas12a, Cas12b, and Cas12e[20,29–33] (Fig. 1a–e and Supplementary Fig. 1a-d). In Cas12i1, the REC lobe consists of the Helical-I domain and the PAM-interacting (PI) domain, whereas the NUC lobe comprises the Wedge (WED), Helical-II, RuvC, and Nuc domains (Fig. 1a, d and Supplementary Fig. 2a-g). The mature crRNA and target DNA are sandwiched inside the positively charged pocket formed by the two lobes (Fig. 1b–e). To be noted, two separate α-helix bundles in the Helical-I domain are located at the PAM-proximal and PAM-distal regions, respectively, which allows the Helical-I domain to extend over the long axis of the whole ternary complex (Supplementary Fig. 2a, b).

A mature 43-nt crRNA consisting of 23-nt repeat region and 20-nt guide region was determined in both ternary complexes. The crRNA originated from a co-expressed CRISPR array template and was processed by the expressed Cas12i1 in E. Coli cells. Overall, the crRNA repeat region approximately adopts a stem-loop structure, and the R-loop formed by crRNA and target DNA in the pre-cleavage R-loop complex displays a T-shaped architecture (Fig. 1c and Supplementary Fig. 3a). The nucleotides U(1)–A(19) of the crRNA guide region and dA(1)–dT(19) of the target DNA (TD) strand form a 19-bp heteroduplex. The nucleotides dG(−1)–dC(−9) of the TD strand and dC(−1*)–dG(−9*) of the non-target DNA (NTD) strand form the PAM duplex (Fig. 1b, c).

**Recognition of the crRNA repeat region.** The extensive interactions between the stem-loop structure of the crRNA repeat region and Cas12i1 are mainly involved in stabilizing the sugar-phosphate backbone of the stem-loop structure (Supplementary Figs. 4a, b and 5). The short helix α2 of the WED domain is located above the G(−17):C(−1) base pair of the stem duplex, which markedly distorts orientations of the nucleotides U(−19) and U(−18) (Supplementary Fig. 4c). Consequently, the 5′ unpaired repeat region nucleotides A(−23)–U(−18) swing away from the stem duplex and extend into the pre-crRNA processing pocket within the WED domain. Particularly, the nucleotide A(−23) stacks with residues H497 and H528 (Supplementary Fig. 4c). The nucleotide U(−22) stacks with the side chain of R503 and forms hydrogen bonds with K494 and D507. The nucleotide U(−21) stacks with Y509 and hydrogen bonds with W511. The nucleotide U(−18) hydrogen bonds with Y509 and the backbone oxygen atom of G584. Details of the interactions between Cas12i1 and the crRNA are presented in Supplementary Fig. 5.

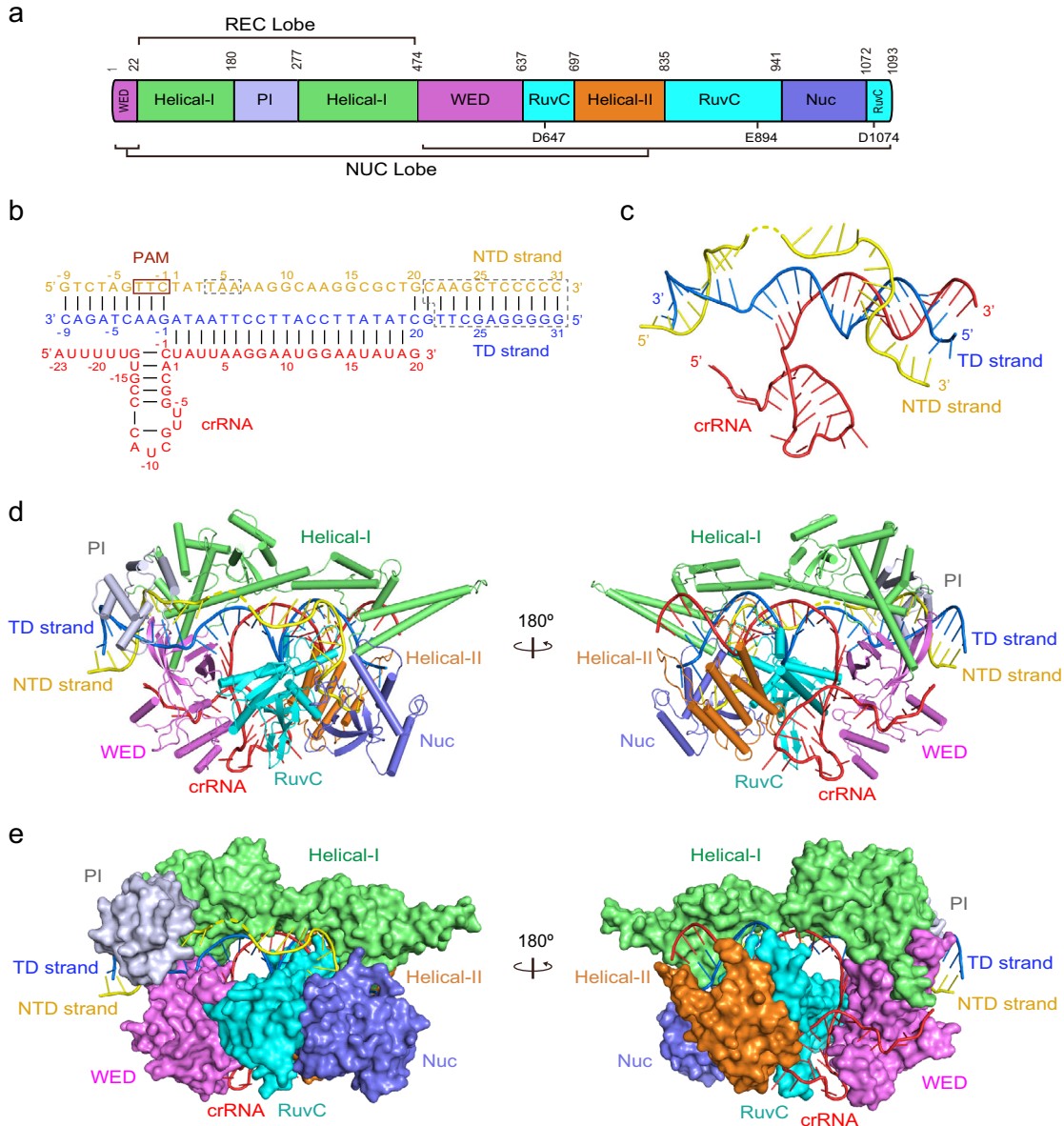

**Fig. 1 Crystal structure of the Cas12i1 pre-cleavage R-loop complex. a** Domain organization of Cas12i1. Catalytic residues within the RuvC domain are annotated below the diagram. **b** Schematic representation of the R-loop structure formed by crRNA and the target duplex. Disordered regions are surrounded by gray dashed lines. **c** View of the R-loop structure in the complex. **d** Overall structure of the Cas12i1 pre-cleavage R-loop complex shown in two different orientations and color-coded as defined in (**a**) and (**b**). **e** Surface representations of the Cas12i1 pre-cleavage R-loop complex shown in the same views as in (**d**).

**Mechanism of the 5′-TTN-3′ PAM determination**. The PAM duplex is gripped inside a positively charged cleft formed by the Helical-I, PI, and WED domains (Fig. 2a, b). The PI domain is connected to the Helical-I domain via two loops, which would provide the PI domain with remarkable flexibility prior to binding the PAM duplex. In pre-cleavage R-loop complex, a loop between the helices α3 and α4 of the PI domain inserts into the minor groove of the PAM duplex to contact the PAM nucleotides dT (−3*) and dT(−2*) in the NTD strand (Fig. 2a). Concurrently, another loop between the strands β2 and β3 of the WED domain inserts into the major groove of the PAM duplex and interacts with the PAM-complementary nucleotides dA(−3)–dG(−1) in the TD strand. The 5′-TTN-3′ PAM is recognized mainly by hydrogen bonds. Firstly, the O4 atom of dT(−3*) hydrogen bonds with the side chain of H170 and the 5-methyl group of dT(−3*) engages in the van der Waals interaction with the side chain of

L298 (Fig. 2c). The N6 and N7 atoms of dA(−3) form hydrogen bonds with the side chain of N481 (Fig. 2d). Notably, the Cα atom of residue G235 is adjacent to the nucleobases of the dA(−3):dT (−3*) base pair, suggesting that any substitution of G235 would directly disrupt the correct recognition of this base pair. Modeling of the dG(−3):dC(−3*) base pair to replace the dA(−3):dT(−3*) base pair would generate a steric clash between the N2 atom of dG (−3) and the backbone nitrogen atom of residue A236 (Supplementary Fig. 6a). Secondly, the O2 atom of dT(−2*) hydrogen bonds with the backbone nitrogen atom of A236 (Fig. 2c). The N6 and N7 atoms of dA(−2) form hydrogen bonds with the side chain of S482 (Fig. 2d). Modeling of the dG(−2):dC(−2*) base pair to replace the dA(−2):dT(−2*) base pair would also generate steric clashes between the N2 atom of dG(−2) and the side chain of A236 (Supplementary Fig. 6b), thus explaining the requirement of double T in the PAM. Thirdly, the O6 atom of dG(−1) forms a

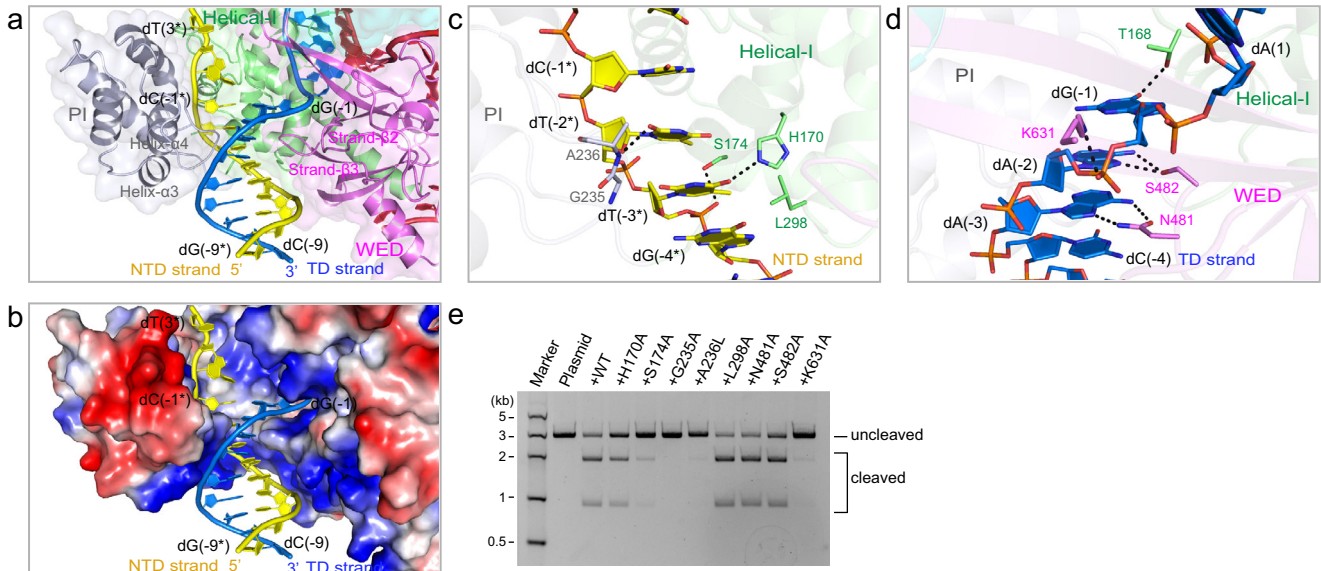

**Fig. 2 Recognition of the PAM duplex. a** Recognition of the PAM duplex in the Cas12i1 pre-cleavage R-loop complex and the PAM-interacting cleft formed by the Helical-I, PI, and WED domains. The loop between strands β2 and β3 of the WED domain inserts into the major groove of the PAM duplex, and another loop between helices α3 and α4 in the PI domain inserts into the minor groove of the PAM duplex. **b** Electrostatic potential surface of the PAM-interacting cleft. Red, white, and blue indicate negative, neutral, and positive electrostatic potential surfaces, respectively. **c** Recognition of the NTD strand in the PAM duplex. **d** Recognition of the TD strand in the PAM duplex. **e** Agarose gel demonstrating the cleavage of the linearized plasmid by wild-type Cas12i1 and the mutants in complex with crRNA. The cleavage assays were repeated three times independently to confirm the repeatability and source data are provided as a Source Data file.

hydrogen bond with the side chain of T168 (Fig. 2d). It lacks specific recognition at this position in the PAM. Aside from the aforementioned interactions, a number of other Cas12i1 residues play a role in stabilizing the sugar-phosphate backbone of the PAM duplex (Supplementary Fig. 5). The results of the cleavage assays show that the previously unreported G235A and A236L mutations almost abolished the cleavage activity. The S174A and K631A mutations decrease this activity (Fig. 2e). Therefore, residues G235 and A236 play a critical role in PAM determination.

**Mechanism of the R-loop formation in Cas12i1.** In the pre-cleavage R-loop complex, the two strands of target DNA are unwound beyond the PAM duplex and the TD strand forms a 19-bp heteroduplex with the crRNA guide region. The PAM-proximal region of the heteroduplex contacts the Helical-I, WED, and RuvC domains, and the PAM-distal region of the hetero-duplex interacts with the Helical-I, Helical-II, and RuvC domains (Fig. 1d). The NTD strand extends from the PAM duplex and is partially exposed to the surrounding solvent. The nucleotides dT (4*)–dA(6*) of the NTD strand are disordered and invisible due to the flexibility, and the nucleotides dA(7*)–dG(20*) pass through the channel formed by the Helical-I, Helical-II, RuvC, and Nuc domains. Finally, the nucleotide dG(20*) of the NTD strand interacts with dC(20) of the TD strand (Fig. 1c, d).

Notably, helix α7 and a loop region between helices α6 and α7 of the Helical-I domain are positioned to obstruct the possible dA (1)–dA(3):dT(1*)–dT(3*) base pairings and prompt the flipping of dA(1)–dA(3), thus behaving as a "zipper" for unwinding of the target DNA duplex, and are henceforth referred to as the zipper motif (residues K160–F177) (Fig. 3a). To investigate the function of the zipper motif, we generated a protein construct in which the loop region (residues K160–A169) of the zipper motif was replaced by a simple linker (residues GGSGGS). When cleaving DNA substrates containing 1–4-bp mismatched bubbles adjacent to the PAM, the △160–169aa construct show the decreased cleaving capacity (Fig. 3b). The cleavage activity was restored by

increasing the length of mismatched bubbles, which denotes that the zipper motif plays an important role for facilitating the target duplex unwinding. Therefore, binding of the target duplex and subsequent recognition of the correct PAM sequence by Cas12i1 could make the PAM-proximal region of the target duplex approach the zipper motif, and the zipper motif could generate a spatial obstruction toward base pairing within the PAM-proximal region of the target duplex and contribute to the nucleation between the TD strand and the seed region of the crRNA guide.

Besides the zipper motif, other regions or residues of Cas12i1 also play a role in the target duplex unwinding. Helix α2 and strand β1 of the WED domain are arranged above the stem duplex of the crRNA repeat region (Fig. 3c). They mediate a flipping of the nucleotide U(1) relative to C(−1) in the crRNA and facilitate base pairing between U(1) of the crRNA and dA(1) of the TD strand. More specifically, the side chain of R535 stacks with the pyrimidine ring of C(−1), while residues R535 and K923 form salt bridges with the phosphate group between U(1) and C(−1), which profoundly shapes the orientation of U(1) in the crRNA. Residues R535 and K13 interact with E539 to further stabilize the orientation of U(1). Concurrently, the side chain of R12 stacks with the purine ring of dA(1) in the TD strand, and residues R12 and K483 form salt bridges with the phosphate group between dA (1) and dG(−1). These interactions facilitate the base pairing between dA(1) of the TD strand and U(1) of the crRNA. Such kind of "phosphate lock" interactions has also been found in Cas9, Cas12a, and Cas12b[29,33–37]. The mutational studies show that the R535A mutation greatly decreases the cleavage activity, and the R12A, K483A, and K13A mutations slightly decrease this activity (Fig. 3d). Altogether, two groups of "phosphate lock" interactions, one group (residues R535 and K923) interacting with the crRNA and the other group (residues R12 and K483) interacting with the TD strand, have been identified in Cas12i1 to facilitate the nucleation between the TD strand and the crRNA guide.

The crRNA guide-TD strand heteroduplex is flanked by Cas12i1 on both sides throughout its length. The long helix α9

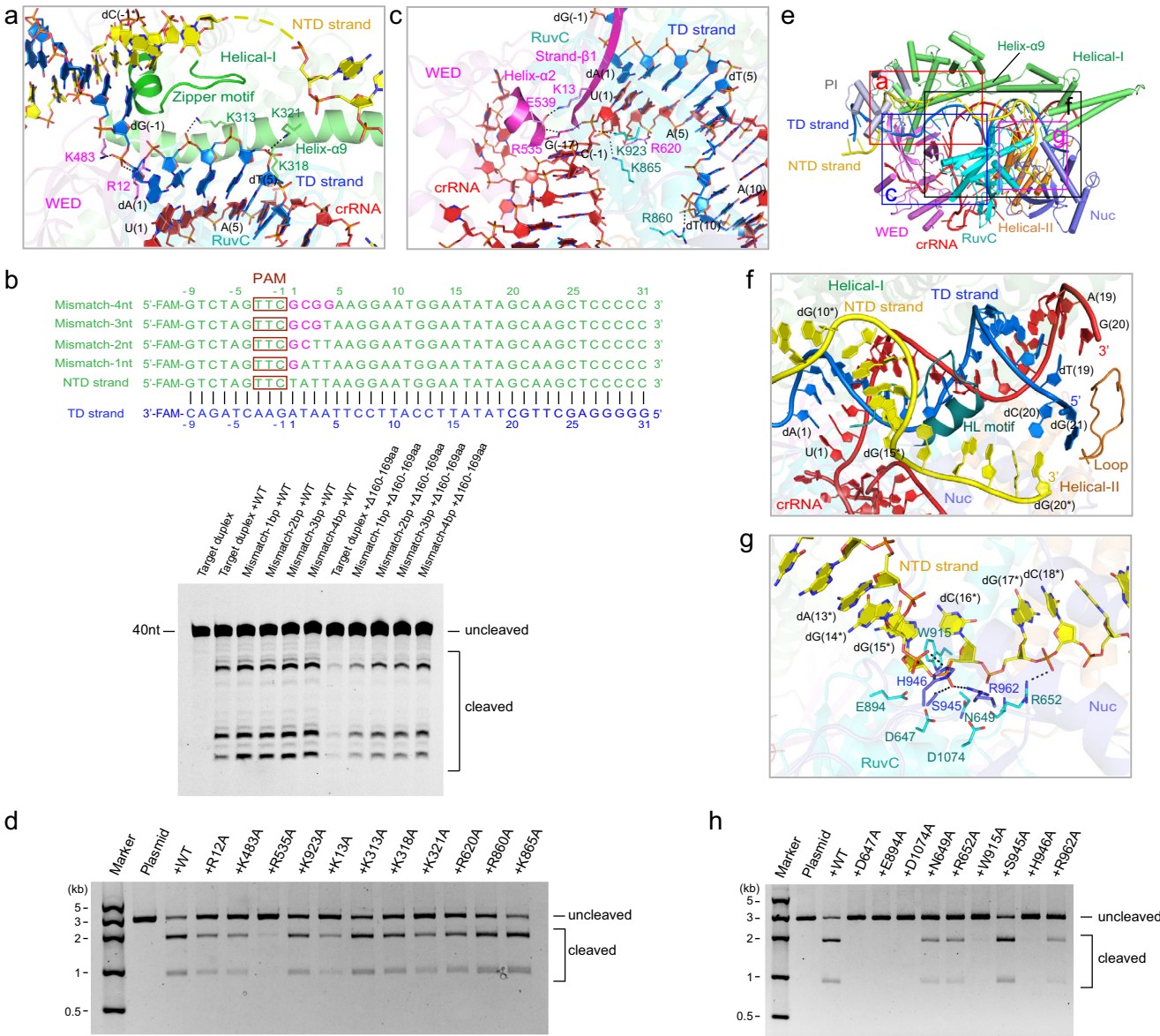

**Fig. 3 The target duplex unwinding and R-loop formation. a** Detailed interactions between the unzipped target duplex and Cas12i1 at the PAM-proximal region of the Cas12i1 pre-cleavage R-loop complex. The zipper motif (residues 160–177aa) facilitates unwinding of the target duplex. **b** Schematic representations of the fluorescently labeled target duplex and 1–4-nt mismatched NTD strands designed to form 1–4-bp mismatched duplexes. Denaturing gel demonstrating the cleavage of the fluorescently labeled target duplex or mismatched duplexes by wild-type Cas12i1 and the △160–169aa mutant in complex with crRNA. **c** Detailed interactions between the PAM-proximal region of the heteroduplex and Cas12i1. **d** Agarose gel demonstrating the cleavage of the linearized plasmid by wild-type Cas12i1 and the mutants in complex with crRNA. **e** The relative positions of (**a**), (**c**), (**f**), and (**g**) in the Cas12i1 pre-cleavage R-loop complex. The long helix α9 of the Helical-I domain is labeled. **f** The PAM-distal region of the R-loop structure. The HL motif (residues 895–915aa) of the RuvC domain is located between the crRNA guide region-TD strand heteroduplex and the unpaired NTD strand. A loop region of the Helical-II domain helps the TD strand interacting with the NTD strand. **g** Detailed interactions between the unpaired NTD strand and the RuvC catalytic pocket of Cas12i1. Residue A647 of the catalytically inactive Cas12i1 pre-cleavage R-loop complex was virtually mutated back to D647. **h** Agarose gel demonstrating the cleavage of the linearized plasmid by wild-type Cas12i1 and the mutants related to the RuvC catalytic pocket in complex with crRNA. The cleavage assays of (**b**), (**d**), and (**h**) were repeated three times independently to confirm the repeatability, respectively, and source data are provided as a Source data file.

of the Helical-I domain stretches along one flank of the heteroduplex (Fig. 3a, e). Residues K313, K318, and K321 of helix α9 stabilize the sugar-phosphate backbone of the PAM-proximal region in the TD strand (Fig. 3a). On the other flank of the heteroduplex, a helix-loop (HL) motif (residues N895-W915) in the RuvC domain acts as a guardrail that restricts the unpaired region of the NTD strand and the heteroduplex to two separate channels (Fig. 3f). The heteroduplex-binding channel is formed by the Helical-I, Helical-II, and RuvC domains, whereas the NTD

strand-binding channel is formed by the Helical-I, Helical-II, RuvC, and Nuc domains. Single mutations to alanine of residues K313, K318, K321, R620, R860, and K865 interacting with the heteroduplex have minor effects on the cleavage activity (Fig. 3a, c, d). At the PAM-distal end of the pre-cleavage R-loop complex, a loop region (residues N724–I737) between helices α1 and α2 of the Helical-II domain impedes base pairing between dC(20) and G(20) (Fig. 3f). Consequently, the nucleotides dC(20) and dG(21) of the TD strand make a sharp turn relative to dT(19), which

facilitates dC(20) interacting with dG(20*). Due to the flexibility or instability, the target duplex beyond the dC(20):dG(20*) base pair is hardly distinguishable in the present model.

The structure of the pre-cleavage R-loop complex displays how the NTD strand is accommodated within the RuvC active site (Fig. 3g). Residue W915 intervenes between dG(15*) and dC(16*), and affects the orientation of dC(16*). Residue H946 interacts with the phosphate group of dG(15*). W915 and H946 work together to restrict the sugar-phosphate backbone of the NTD strand passing through the RuvC active site. Side chains of three catalytic residues D647 (virtual modeling of A647D), E894, and D1074 in the RuvC domain point toward the phosphate group between the nucleotides dG(15*) and dC(16*), implying this position as the possible cutting site in the NTD strand. Moreover, residues S945 and R962 interact with the phosphate group of dC(16*), and R652 forms a salt bridge with the phosphate group of dC(18*). The results of the cleavage assays show that the D647A, E894A, and D1074A mutants are catalytically inactive, the W915A and H946A mutants display near-complete loss of the cleavage activity, and the N649A, R652A, and R962A mutants have a decreased cleavage activity (Fig. 3h).

**The sequential *cis*-cleavage of the NTD and TD strands**. To investigate the sites at which Cas12i1 cuts two strands of the target duplex, the 40-nt NTD and TD strands were labeled with two different dyes and annealed (Fig. 4a). The cleavage assay shows that the NTD strand was successively cleaved at 13–15 nucleotides after the PAM duplex, whereas the TD strand was cleaved at 24 nucleotides after the PAM duplex (Fig. 4b). Given that the unpaired region of the NTD strand preoccupies the RuvC active site of the pre-cleavage R-loop complex, it is not difficult to understand that Cas12i1 cleaves the NTD strand prior to the TD strand. The sequential *cis*-cleavage of the NTD and TD strands has been observed or discussed in Cas12a, 12b, and 12e effectors[20,29,38,39], and is further validated for Cas12i1 by our time-course cleavage assay (Fig. 4b).

In the post-cleavage R-loop complex, both NTD and TD strands are already cleaved by Cas12i1. The PAM-distal region of the target duplex is released, and the PAM-proximal region of the target duplex is retained in the complex (Fig. 4c, d and Supplementary Fig. 3b). Interestingly, the cleaved NTD strand stretches out of Cas12i1 and is captured by the neighboring complex from another asymmetric unit (Fig. 4e). Correspondingly, the cleaved NTD strand of the neighboring complex is determined passing through the RuvC catalytic pocket of the post-cleavage R-loop complex. The overall conformation of the unpaired region dA(6*)–dA(13*) of the NTD strand from the neighboring complex extended into the post-cleavage R-loop complex resembles that of the unpaired region dC(11*)–dC(18*) of the NTD strand in the pre-cleavage R-loop complex (Fig. 4f). Hence, the post-cleavage R-loop complex not only represents a state of the R-loop complex after the *cis*-cleavage of the target duplex, but also provides additional information about the pre-cleavage state for the *trans*-cleavage.

For some unknown reason, the cleaved NTD strand accommodated within the RuvC active site was not further *trans*-processed in the post-cleavage R-loop complex. Notably, three water molecules were identified to form multiple hydrogen bonds within the RuvC catalytic site (Fig. 4g). Among them, one water molecule interacts with residues D647, N649, D1074 and the phosphate group of dC(11*). Another water molecule forms hydrogen bonds with residues D647, E894, T944 and S945. Typically, the conserved RNase H fold utilizes a two-metal-ion catalytic mechanism for phosphoryl hydrolysis[40,41]. Considering that the RuvC domain in Cas12i1 also possesses such RNase H

fold, positions of these water molecules in the present structure are possibly occupied by two-metal ions, which would be in accordance with the recently reported presence of two metal ions in the RuvC active site of the Cas12i2 ternary complex[28].

**A loop region in the Helical-II domain facilitates the TD strand loading into the RuvC active site**. Following the cleavage of the NTD strand, the mechanism of the TD strand loading into the RuvC active site for cleavage remains to be clarified. The TD strand should enter the RuvC active site via a pathway surrounded by the Helical-II, RuvC, and Nuc domains. In the pre-cleavage R-loop complex, a loop region in the Helical-II domain facilitates bending of the 5′-terminal of the TD strand toward the Nuc domain, in which the TD strand interacts again with the NTD strand (Fig. 5a). We substituted this loop region (residues N724–I737) or the region containing this loop and two flanking helices (residues L706–E750) with a linker (residues GGSG). The cleavage assays show that both constructs affect the *trans*-cleavage pattern and slightly increase the *trans*-cleavage activity for the NTD strand, whereas they decrease the *cis*-cleavage activity for the TD strand (Fig. 5b). Therefore, this loop region in the Helical-II domain facilitates the loading of the TD strand beyond the heteroduplex into the RuvC active site for cleavage.

**Base pairings within the seed region mimicking the target interrogation**. To understand how Cas12i1 recognizes mature crRNA and the conformational features prior to the target duplex binding, we solved the crystal structure of the Cas12i1-crRNA binary complex at 3.6 Å resolution. The global structure of Cas12i1 in the binary complex also exhibits the bilobed architecture, and the crRNA repeat region is recognized by Cas12i1 in the manner observed in the R-loop complexes (Fig. 6a). Nucleotides U(1)–A(15) of the crRNA guide region were identified in the binary complex. The first six nucleotides U(1)–U(6) adopt a preorganized and nearly A-form helical conformation (Fig. 6b). In Cas9, Cas12a, and Cas12b, the preorganized PAM-adjacent seed region within the crRNA guide region is essential for target recognition[29,42,43]. The seed region of the Cas12i1 crRNA is approximately within the first seven nucleotides on the 5′ end of the crRNA guide region[27]. Interestingly, the seed nucleotides A(2)–U(4) in the present study pair with a complementary 3-nt oligonucleotide that originated from *E. coli* cells and was captured by the binary complex (Fig. 6a, b and Supplementary Fig. 3c). Besides, the other seed nucleotides U(1) and A(5)–A(6) maintain a preordered conformation. These base pairings within the seed region mimic the scene of target interrogation by the Cas12i1-crRNA interference module.

**Target binding activates the Cas12i1 DNase activity**. In the binary complex, the PI domain protrudes from the main body of Cas12i1 and extends into the surrounding solvent (Fig. 6a). Due to the flexibility, a part of the PI domain including helix α3 is invisible. An overlay of the binary and ternary complexes reveals that the PAM-interacting cleft undergoes an "open to closed" conformational transition to clench the PAM duplex (Fig. 6c–e). Inside the PI domain, binding to the PAM duplex induces conformational adjustment of helices α2 and α4 relative to helices α1 and α5. Thus, two key features of the resting state of the Cas12i1-crRNA binary complex that poise it for the target duplex binding are the open PAM-interacting cleft and preorganized seed region.

The overlay also indicates that the Helical-I domain exhibits an apparent rearrangement, whereas the NUC lobe as a whole makes a slight rearrangement (Fig. 6c, d). In the binary complex, the RuvC active site is shielded from substrate binding by the PAM-distal α-helix bundle of the Helical-I domain and the HL motif of the RuvC domain (Fig. 6f, g). Besides, Cas12i1 arranges its

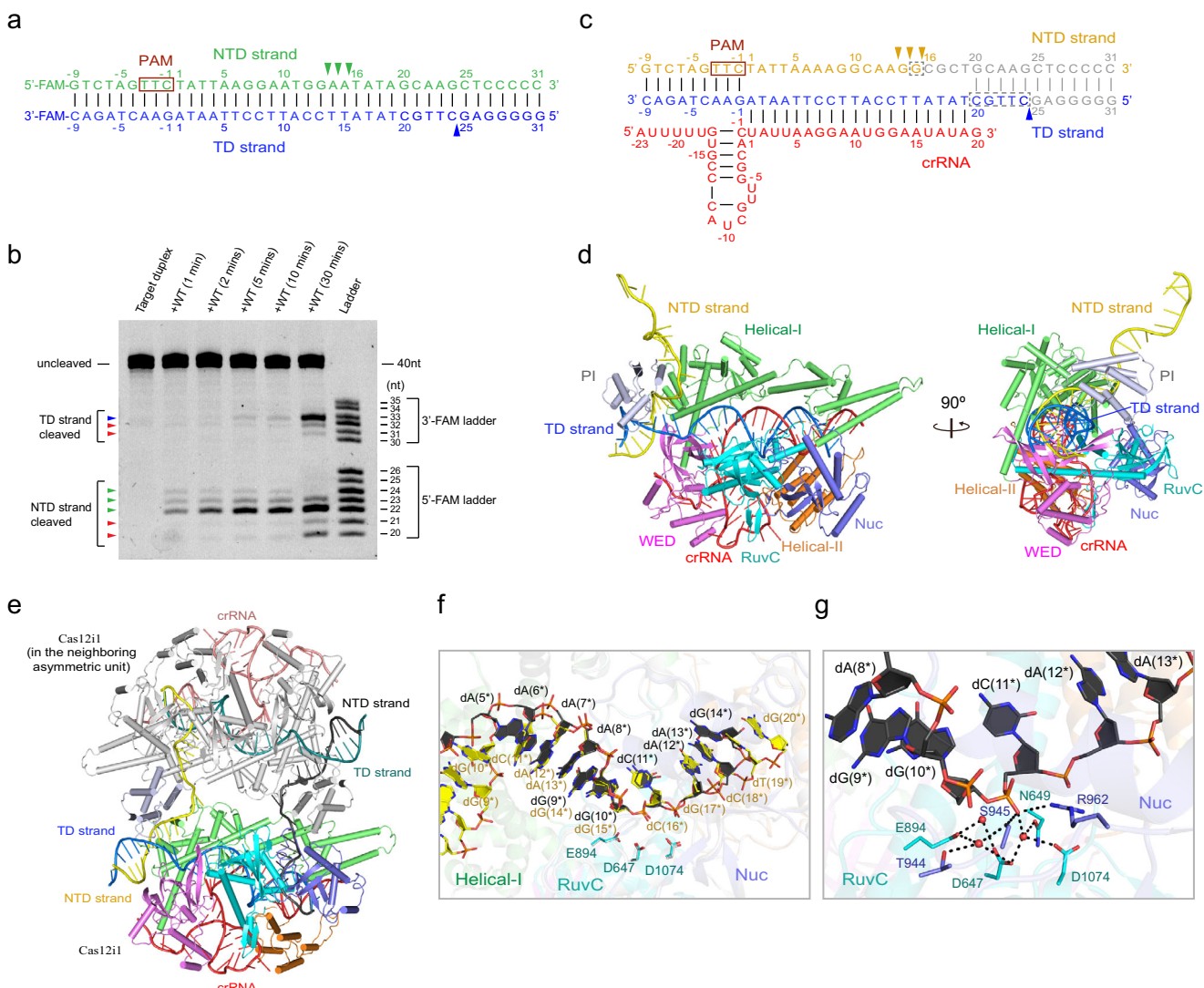

**Fig. 4 Crystal structure of the Cas12i1 post-cleavage R-loop complex. a** Schematic representations of the fluorescently labeled target duplex used in the cleavage assays. Green and blue triangles indicate the cleavage sites within the NTD and TD strands, respectively. **b** Denaturing gel demonstrating the time-course cleavage of the fluorescently labeled target duplex by wild-type Cas12i1 in complex with crRNA. The *cis*-cleavages of the NTD and TD strands are labeled with the green and blue triangles, respectively, and the *trans*-cleavages are labeled with the red triangles. The cleavage assays were repeated three times independently to confirm the repeatability and source data are provided as a Source data file. **c** Schematic representation of the R-loop structure after target DNA cleavage. Yellow and blue triangles indicate the cleavage sites within the NTD and TD strands, respectively. Nucleotides colored in gray represent the PAM-distal cleavage product that was released from the complex. Disordered regions are encircled by gray dashed lines. **d** Overall structure of the Cas12i1 post-cleavage R-loop complex shown in two different orientations, color-coded as defined in Fig. 1. **e** The cleaved NTD strand of the Cas12i1 post-cleavage R-loop complex is extended into the RuvC active site of the neighboring complex from another asymmetric unit and vice versa. Each asymmetric unit contains one post-cleavage R-loop complex. **f**, The overlay of the unpaired NTD strand from the pre-cleavage R-loop complex and the unpaired NTD strand from the neighboring complex extended into the post-cleavage R-loop complex. The pre-cleavage R-loop complex is colored as in Fig. 1. The neighboring unpaired NTD strand and Cas12i1 in the post-cleavage R-loop complex are colored black and silver, respectively. **g** Detailed interactions between the neighboring unpaired NTD strand and the RuvC active site of the post-cleavage R-loop complex. The figure is colored as in (**e**).

multiple domains to form one central channel for binding the crRNA guide region. During transition to the ternary complex, Cas12i1 rearranges the Helical-I domain and the HL motif to form two separate channels, one for the heteroduplex and the other for the NTD strand or the *trans*-cleavage substrate (Fig. 6f, g). Notably, the overlay shows that the target duplex binding drives upward movement of the PAM-distal α-helix bundle of the Helical-I domain (Fig. 6c, d). Moreover, the HL motif of the RuvC domain swings toward the center of Cas12i1 (Fig. 6f, g). As a result of these pivotal changes, the PAM-distal α-helix bundle and the HL motif no longer obstruct the RuvC active site and the DNase activity of Cas12i1 is unleashed.

To understand how many base pairings between the crRNA guide region and the TD strand are required for the conformational activation of Cas12i1, we use double-stranded (ds) or single-stranded (ss) target DNA with different lengths to perform the cleavage assays. The results indicate that a 12-bp target dsDNA and a 12-nt target ssDNA can achieve detectable cleavage activity, whereas 14–16-bp target dsDNA and 13–16-nt target ssDNA can fully activate the enzyme (Fig. 6h).

**The temperature and metal-ion dependent substrate cleavage.** To investigate whether the substrate cleavage by Cas12i1 is a

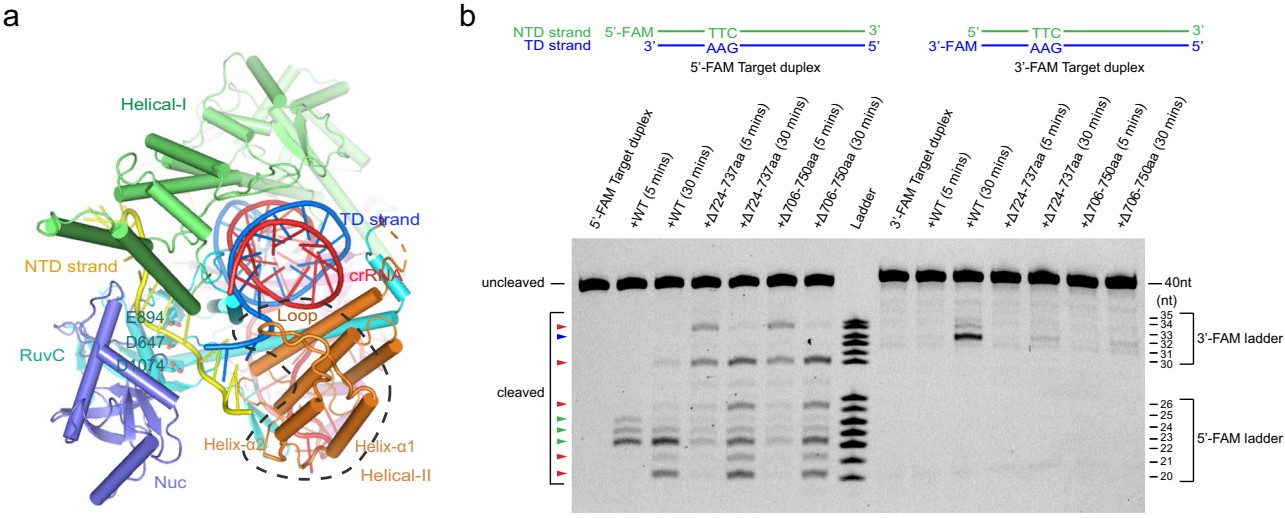

**Fig. 5 A loop region in the Helical-II domain facilitating the TD strand loading. a** A loop region (residues 724–737aa) and the region comprising this loop and two connected helices (residues 706–750aa) in the Cas12i1 R-loop ternary complex are encircled with a black dashed line. **b** Denaturing gel demonstrating the cleavage of the fluorescently 5′-FAM-NTD strand-labeled or 3′-FAM-TD strand-labeled DNA substrate by wild-type Cas12i1, △724–737aa, and △706–750aa mutants in complex with crRNA. The *cis*-cleavages of the NTD and TD strands are labeled with the green and blue triangles, respectively, and the *trans*-cleavages are labeled with the red triangles. The cleavage assays were repeated three times independently to confirm the repeatability and source data are provided as a Source data file.

temperature or metal-ion-dependent process, we carried out the cleavage assays at different temperatures or by adding different metal ions to the reaction mixture. The results show that Cas12i1 is capable of cleaving substrates between 37 and 57 °C, and has the favorable cleaving temperature of 45 °C (Supplementary Fig. 7a). Moreover, among the tested divalent metal ions, only $Mg^{2+}$ ion is essential for the substrate cleavage by Cas12i1 (Supplementary Fig. 7b).

## Discussion

In the recently reported cryo-EM and crystal structures of Cas12i1 and Cas12i2 complexes, the partial target duplex containing the PAM sequences was presented[27,28]. The mechanism by which Cas12i unzips the double-stranded target duplex was short of investigation and information related to the R-loop formation and recognition in Cas12i was almost not reported. To address these questions, we report crystal structures of Cas12i1 ternary complexes in the pre-cleavage and post-cleavage R-loop states. The zipper motif and two groups of "phosphate lock" interactions have been identified to facilitate the target DNA duplex unwinding in Cas12i1. In both Cas12i1 R-loop complexes, the TD strand is unwound from the NTD strand beyond the PAM duplex and forms a 19-bp heteroduplex with the crRNA guide region. In the Cas12i1 pre-cleavage R-loop complex, the nucleotide of the TD strand swings to interact with that of the NTD strand beyond the 19-bp heteroduplex. In contrast, 28-bp and 26-bp heteroduplexes were previously reported to be accommodated in the Cas12i1 and Cas12i2 ternary complexes, respectively[27,28], possibly due to the incomplete states of the target DNA duplexes and longer crRNA guide regions used to prepare the ternary complexes. This implies that the complete R-loop structure in Cas12i will affect the reasonable length of the heteroduplex formed by the TD strand and the crRNA guide region.

Cas12i1 and Cas12i2 share 27.25% protein sequence identity. The overlay of the Cas12i1 and Cas12i2 ternary complexes clearly shows that two proteins have similar structural folds (Supplementary Fig. 8a-c). The Helical-I domain in Cas12i1 contains two separate α-helix bundles, and their counterparts in Cas12i2 are

defined as two independent domains (the Helical-I and Helical-II domains), which is the main difference between domain compositions of Cas12i1 and Cas12i2. Although nucleotide sequences of the crRNA repeat regions in Cas12i1 and Cas12i2 systems are obviously different, the stem-loop architectures for their crRNA repeat regions resemble each other closely (Supplementary Fig. 8a-e). In the present study, we proved that Cas12i1 cleaved the NTD strand prior to the TD strand. The NTD strand was successively cleaved 13–15 nucleotides after the PAM duplex, and the TD strand was cleaved 24 nucleotides after the PAM duplex. Cas12i2 was also found to cleave the TD strand at the same position as observed in Cas12i1, whereas the NTD strand was primarily cleaved 31 nucleotides after the PAM duplex by Cas12i2[28]. In addition, the post-cleavage R-loop complex of Cas12i1 provides not only the information of the R-loop structure after the *cis*-cleavage of the target duplex, but also the information regarding the Cas12i1 *trans*-cleavage.

Due to the lack of the PI domain in the previously reported binary complexes of Cas12i1 and Cas12i2, the conformational movement of the PI domain was not observed[27,28]. This study noticed a remarkable "open to closed" conformational transformation of the PI domain upon the target duplex binding in Cas12i1 (Fig. 6c–e). We also found that residues G235 and A236 of the Cas12i1 PI domain played critical roles in the 5′-TTN-3′ PAM determination. In addition, the previous study revealed that the HL motif (also termed the Lid motif) of Cas12i1 underwent a conformational rearrangement during transition from binary to ternary complexes[27]. We found that both the HL motif and the PAM-distal α-helix bundle of the Helical-I domain in Cas12i1 were significantly rearranged so that they no longer obstructed the substrate accessing to the RuvC active site and the Cas12i1 DNase activity was unleashed. Such conformational rearrangement of the Helical-II domain of Cas12i2 was also reported to activate the Cas12i2 DNase activity[28].

Although Cas12a, Cas12b, Cas12e, and Cas12i possess the REC and NUC bilobed architectures, the composition and domain distribution of these effectors display considerable divergence[20,29–33,37–39,43,44]. Cas12i1 shares most structural features with Cas12b (Supplementary Fig. 1a, b). For instance, the Helical-I domains of Cas12i1 and Cas12b

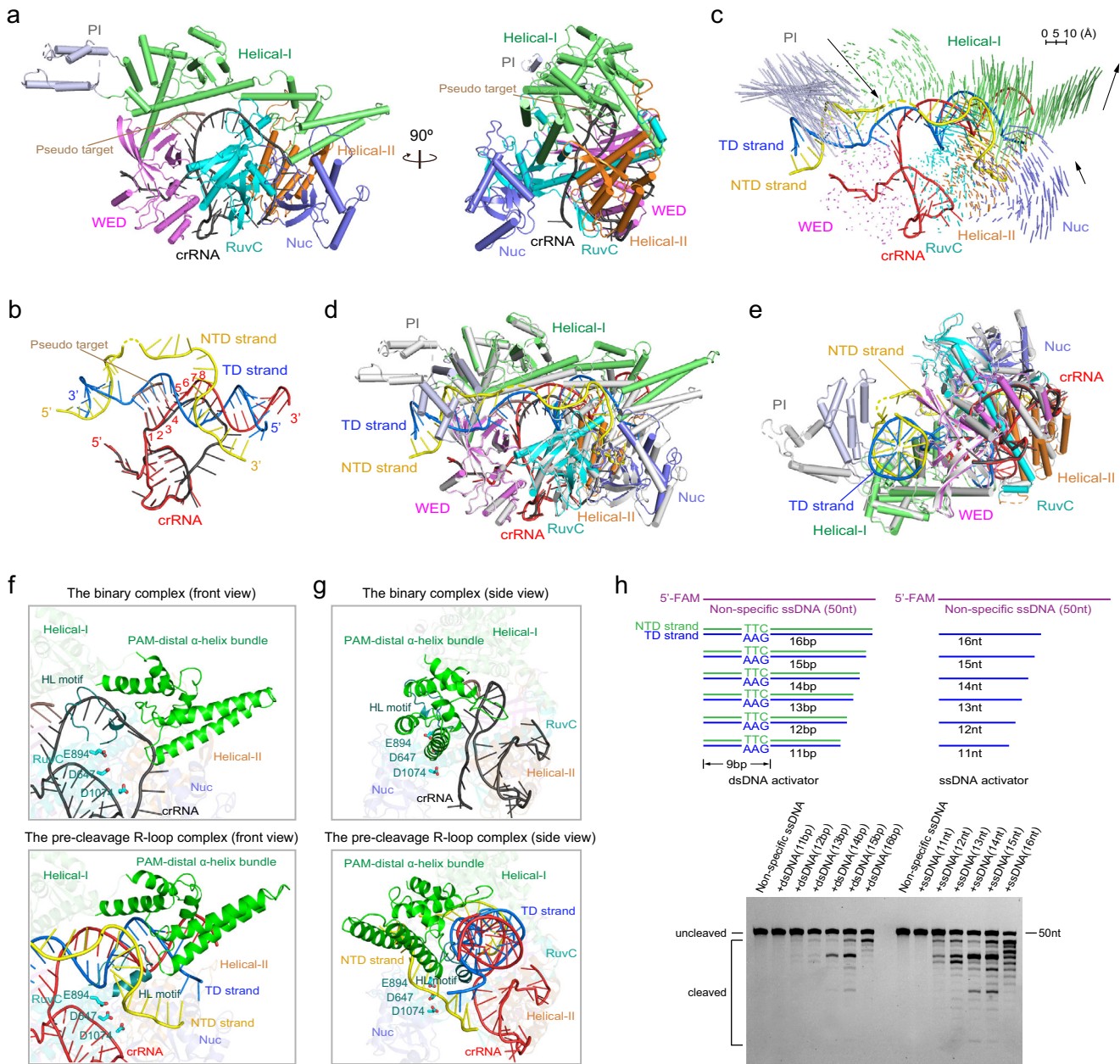

**Fig. 6 The target binding triggers the Cas12i1 DNase activity. a** Overall structure of the Cas12i1 binary complex shown in two different orientations. **b** Structural comparison of crRNAs in the Cas12i1 binary and pre-cleavage R-loop complexes. crRNAs are colored as in (**a**) and Fig. 1. **c** Structural comparison of Cas12i1 in the binary and pre-cleavage R-loop complexes. Vector length correlates with the motion scale of each domain in Cas12i1. Black arrows indicate domain movements upon the target duplex binding. **d**, **e** The overlay of the Cas12i1 binary and pre-cleavage R-loop complexes shown in front and side views. Cas12i1 and crRNA in the binary complex are colored in silver and black, and the pre-cleavage R-loop complex is colored as in Fig. 1. Note that the PI and Helical-I domains undergo significant rearrangements. **f** Close-up front views of the Cas12i1 binary and pre-cleavage R-loop complexes. The PAM-distal α-helix bundle of the Helical-I domain exhibits an upward movement and the HL motif rearranges its conformation. **g** Close-up side views of the Cas12i1 binary and pre-cleavage R-loop complexes. A single central channel can be observed in the binary complex to accommodate the crRNA guide region, whereas two channels separated by the HL motif are formed in the pre-cleavage R-loop complex to accommodate the crRNA guide region-TD strand heteroduplex and the NTD strand. **h** Schematic representations of double-stranded DNA (dsDNA) activators and single-stranded DNA (ssDNA) activators with different lengths. Denaturing gel demonstrating the cleavage of the fluorescently labeled non-specific ssDNA by wild-type Cas12i1 in complex with crRNA and in the presence of dsDNA or ssDNA activator. The cleavage assays were repeated three times independently to confirm the repeatability and source data are provided as a Source data file.

have a similar structural fold. The Cas12i1 PI domain and the Aac-Cas12b (*Alicyclobacillus acidoterrestris* Cas12b) counterpart (residues 136–173aa) are connected to the Helical-I domains of the respective effectors. In contrast, the Cas12a PI domain is located on the opposite side of the PAM duplex and is connected to the WED domain (Supplementary Fig. 1c). Cas12e lacks an equivalent of the PI domain,

but contains a unique non-target strand-binding (NTSB) domain connected to the Helical-I domain (Supplementary Fig. 1d). Moreover, the Helical-II domains of Cas12i1 and Cas12b are connected to the respective RuvC domains, whereas the corresponding Helical-II domains of Cas12a and Cas12e are connected to the Helical-I and WED domains, but not the RuvC domain (Supplementary Fig. 1a-d).

In addition, the crRNA repeat regions in the Cas12i1 and Cas12i2 complexes display a simple stem-loop structure, and the one in the Cas12a complex folds into a pseudoknot structure. Single guide RNAs (sgRNAs) in the Cas12b and Cas12e ternary complexes present distinct and complicated architectures.

In Cas12i1, the zipper motif and the phosphate locks facilitate the target duplex unwinding beyond the PAM duplex, after which the preorganized crRNA seed region contributes to base pairing with the unwound TD strand. We can discern a structural motif (residues 108–130aa) of AacCas12b resembling the zipper motif of Cas12i1, implying that AacCas12b may employ similar mechanism to aid in the target duplex unwinding (Supplementary Fig. 1e). In contrast, FnCas12a (*Francisella novicida* Cas12a) uses a loop-lysine helix-loop (LKL) motif (residues 662–679aa) from its PI domain to promote the target duplex unwinding[44] (Supplementary Fig. 1f). Cas12e instead uses the unique NTSB domain for this purpose[20]. In addition, the phosphate lock and the preorganized seed region have also been observed in Cas12a and Cas12b to facilitate the target duplex unwinding and promote the heteroduplex formation[29,33,37,43]. Following the initial unzipping of the target duplex, the progressive formation of the crRNA guide region-TD strand heteroduplex can greatly reshape conformation of Cas12 effectors. The 19-bp heteroduplex is presented in two Cas12i1 R-loop complexes. The 20-bp heteroduplex has been found in the ternary complexes of Cas12a and Cas12b[29,33,37]. Beyond the 20-bp heteroduplex, the TD strand makes a sharp turn and interacts with the NTD strand in Cas12a[39,43].

In the binary complexes, Cas12 effectors possess a central channel which binds the guide region of their crRNA or sgRNA. Target duplex binding induces significant conformational changes in Cas12 effectors that result in replacement of the crRNA guide region-binding central channel with two separate channels to accommodate the crRNA guide region-TD strand heteroduplex and the unpaired NTD strand. In Cas12a, the PI domain and the REC lobe (including the Helical-I and Helical-II domains) undergo remarkable rearrangements (Supplementary Fig. 9a). The PAM-interacting cleft of Cas12a adjusts from open to closed conformation to bind the target duplex[37]. The Cas12a Helical-II domain detaches from the surface of the RuvC and Nuc domains, leaving the passage to the RuvC active site unobstructed. During the hybridization of the crRNA guide region and the TD strand in Cas12a, certain key motifs such as the finger, helix-loop-helix (HLH), and REC linker from the REC lobe as well as the lid motif in the RuvC domain work concertedly to conformationally activate the DNase activity of Cas12a[38,44]. In AacCas12b, while the REC lobe goes through modest rearrangement[29] (Supplementary Fig. 9b), crucial movement of the HL motif (residues 850–870aa) from the RuvC domain could trigger the DNase activity (Supplementary Fig. 9c). Compared to Cas12a, the PAM-interacting cleft of Cas12i1 undergoes a greater movement to fulfill the open-to-closed adjustment (Fig. 6c–e). Notably, conformational rearrangements of the PAM-distal α-helix bundle from the Helical-I domain and the HL motif from the RuvC domain allow substrates access to the RuvC active site in Cas12i1 (Fig. 6f, g).

All reported Cas12 effectors generate staggered double-strand breaks using the single DNase active site in the RuvC domain. Cas12a and Cas12b cleave the NTD strand at 14 and 17 nucleotides after the PAM, respectively[29,44]. Cas12e and Cas12i1 cut the NTD strand at 12–14 and 13–15 nucleotides after the PAM, respectively[20]. A possible explanation for the successive cleaving of the NTD strand by Cas12i1 is that Cas12i1 first cleaves 15 nucleotides after the PAM, after which the remainder of the NTD strand is stretched and fed into the RuvC active site for two more consecutive cuts. Afterward, the NTD strand could

be further *trans*-processed by Cas12i1. The phenomenon of the successive cleaving of the NTD strand by Cas12e and Cas12i1 should be taken into account for improving the genome-editing fidelity.

The state of the TD strand loading into the RuvC active site was first observed in Cas12b[29]. As seen in the Cas12b ternary complex, direct elongation of the TD strand beyond the heteroduplex would clash with the Helical-II domain of Cas12b (Supplementary Fig. 1b). Consequently, the TD strand is forced to make a reverse turn. Further stabilized and guided by the Nuc domain, the TD strand is loaded into the RuvC catalytic pocket and cleaved at 24 nucleotides after the PAM[29]. The TSL domain in Cas12e and the Nuc domain in Cas12a are speculated to play an active role in unwinding of the PAM-distal target duplex and loading the TD strand into the RuvC catalytic pocket[20,39]. In the Cas12i1 pre-cleavage R-loop complex, we did not observe clear density for nucleotides beyond the 20th nucleotide of the NTD strand after the PAM. A direct elongation of the 3′-end of the NTD strand will clash with helix α2 of the Helical-II domain, which may disturb the PAM-distal base pairing between the TD and NTD strands (Supplementary Fig. 9d). Therefore, the Cas12i1 Helical-II domain may play a key role in unwinding of two strands in the PAM-distal target duplex and facilitates the TD-strand loading into the RuvC catalytic pocket.

In summary, we determine the structures of the Cas12i1 R-loop complexes prior to and after target DNA cleavage, elucidate the mechanisms for the target duplex unwinding, R-loop formation and cleavage of the NTD and TD strands, reveal the role of domains and key components for the TD strand loading into the RuvC active site, and identify critical residues involved in PAM determination. Moreover, we report the structure of the Cas12i1 binary complex, analyze the conformational changes in Cas12i1 that occur during transition from binary to ternary complex, and explain the mechanisms underlying the conformational activation of the Cas12i1 DNase activity. Based on our structural and biochemical studies, we finally propose a multi-step model for the Cas12i1 activation (Fig. 7). Our study provides mechanistic insights for rational application of the CRISPR-Cas12i system and its further development into a powerful tool for research and therapeutic purposes.

## Methods

**Purification of the Cas12i1 binary complex assembled in vivo.** The full-length gene encoding Cas12i1 was codon-optimized and synthesized (Supplementary Table 2). The Cas12i1 gene sequence was amplified by polymerase chain reaction (PCR) and inserted between NdeI and XhoI sites of the pET-30b vector fused with a C-terminal His-tag sequence. CRISPR array template was also designed and synthesized (Supplementary Table 3), after which its gene sequence was cloned into NdeI and XhoI sites of the pCDFDuet-1 vector.

To obtain the Cas12i1 binary complex assembled in vivo, the pET-30b vector encoding the Cas12i1 D647A mutant and the pCDFDuet-1 vector encoding CRISPR template were co-transformed into *E. coli* Rosetta (DE3) cells (Novagen). The bacterial cultures were grown at 37 °C for about 3 h (until OD$_{600}$ reached 0.6), following which 0.2 mM IPTG was added to induce overexpression at 18 °C for 12 h. The cells were collected and lysed in buffer A (25 mM Tris-HCl (pH 7.5), 500 mM NaCl, 3.5 mM β-mercaptoethanol, 1 mM PMSF). In the first step, the binary complex was purified with an Ni-NTA Superflow (QIAGEN) column. The target sample was eluted from the column in buffer A containing gradually increasing concentrations of imidazole. The binary complex sample was further purified with a Heparin HP column (GE Healthcare) using a salt gradient between buffer B1 (25 mM Tris-HCl (pH 7.5), 300 mM NaCl, 2 mM MgCl$_2$, 1 mM DTT) and buffer B2 (25 mM Tris-HCl (pH 7.5), 1.2 M NaCl, 2 mM MgCl$_2$, 1 mM DTT). Afterward, the target sample was dialyzed against buffer C (25 mM Tris-HCl (pH 7.5), 150 mM NaCl, 2 mM MgCl$_2$, 1 mM DTT) and purified with a Superdex 200 Increase (10/300 GL) column (GE Healthcare). The purified sample was concentrated and stored at −80 °C until further use.

The selenomethionine (SeMet)-labeled Cas12i1 (D647A mutant or wild type) binary complexes were expressed in *E. coli* Rosetta (DE3) cells grown in M9 minimal medium supplemented with SeMet, Lys, Phe, Thr, Val, Leu, and Ile. The SeMet-labeled Cas12i1 (D647A mutant or wild type) binary complexes and the

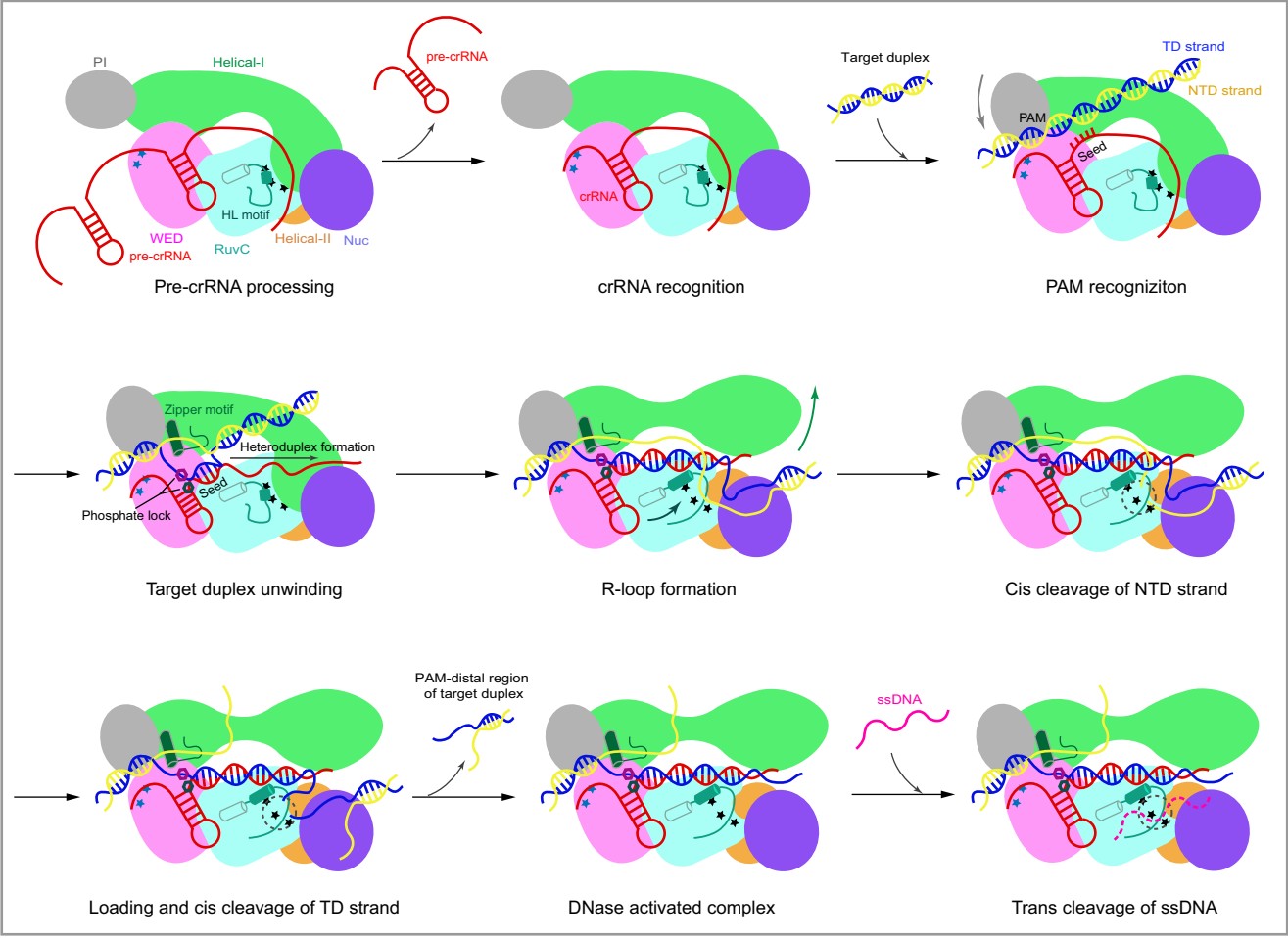

**Fig. 7 A multi-step model for the activation of Cas12i1 *cis*- and *trans*-cleavage.** The apo Cas12i1 recognizes pre-crRNA, processes it into mature crRNA, and forms a Cas12i1-crRNA binary complex. The open PAM-interacting cleft of the binary complex screens the DNA substrate. When matched with a correct PAM, the PAM-interacting cleft of Cas12i1 closes. The zipper motif and the phosphate lock work together to assist unwinding of the target duplex beyond the PAM, and the unwound TD strand is directed toward the preorganized crRNA seed region for interrogation. Correct matching with the crRNA seed region further advances the 19-bp heteroduplex formation, which concomitantly drives conformational rearrangements in Cas12i1. The Helical-I domain and the HL motif are released from the RuvC catalytic pocket to participate in forming of two parallel channels that accommodate the heteroduplex and the unpaired NTD strand, respectively. As a result, the R-loop is formed and the Cas12i1 DNase activity is conformationally activated. Meanwhile, the NTD strand is guided to the RuvC catalytic pocket and cleaved. The Helical-II and Nuc domains then promote unwinding of the PAM-distal target DNA duplex and loading of the TD strand into the RuvC catalytic pocket, following which the TD strand is cleaved and the PAM-distal cleavage product is released from the complex. The activated Cas12i1 could cleave any available single-stranded DNA (ssDNA) in *trans*.

unlabeled Cas12i1 (wild type) binary complex were obtained following the protocol described above.

**Formation of the Cas12i1 ternary complex.** Gel purified 40-nt target strand and non-target strand of target DNA duplex were purchased and dissolved in buffer D (25 mM Tris-HCl (pH 7.5), 100 mM NaCl, 2 mM MgCl₂) to form a 100 mM stock. The two strands were then mixed at a molar ratio of 1:1, denatured at 95 °C for 5 min, and annealed by slow cooling down to room temperature within 1 h. The prepared binary complex and the annealed target duplex were mixed together in buffer C at an approximate molar ratio of 1:1.2 and incubated at 16 °C for 1 h. Following gel-filtration purification with a Superdex 200 Increase (10/300 GL) column (GE Healthcare), fractions containing ternary complex were collected, concentrated to A₂₈₀ = 12.0, and used for further experiments.

**Purification of the uncomplexed Cas12i1.** The wild-type Cas12i1 or its mutant encoded by the pET-30b vector was expressed in *E. coli* Rosetta (DE3) cells (Novagen). After culturing and protein expression, the cells were lysed in buffer A and Cas12i1 was purified using the protocol described for the purification of the binary complex, with the exception that buffers B1, B2, and C did not contain 2 mM MgCl₂. The purified Cas12i1 samples were concentrated and stored for later use in the cleavage assays.

**In vitro synthesis and purification of crRNA.** crRNA sequence was synthesized and purified following the previous protocol[45,46] (Supplementary Table 3). In brief, DNA fragment encoding crRNA was inserted between StuI and HindIII sites of a modified pUC-119 vector. The vector was amplified in *E. coli* DH5α cells, extracted, linearized by HindIII, and purified. In vitro transcription was carried out using homemade T7 RNA polymerase and the reaction sample was resolved using a 20% denaturing (8 M urea) polyacrylamide gel. The target RNA band was excised and eluted using the Elutrap system (GE Healthcare). The eluted RNA sample was concentrated and de-salted by ethanol precipitation, after which it was dissolved in DEPC-treated H₂O and stored at −80 °C.

**Crystallization.** The SeMet-labeled and unlabeled Cas12i1 D647A mutant ternary complexes were crystallized using the hanging-drop vapor diffusion method at 16 °C. Crystals were acquired by mixing 1 μl of complex solution (A₂₈₀ ₙₘ = 12.0) and 1 μl of reservoir solution (0.1 M sodium citrate (pH 5.6), 17% (w/v) Polyethylene glycol 3350, and 0.1 M sodium citrate tribasic dihydrate). Crystals grew to full size within 12 days, and were then harvested and cryoprotected in reservoir solution supplemented with 16% (v/v) glycerol and flash-frozen in liquid nitrogen.

The wild-type Cas12i1 ternary complex was crystallized using the method described above with the reservoir solution adjusted to 0.1 M sodium citrate (pH 5.4), 15% (w/v) Polyethylene glycol 3350, and 0.2 M lithium chloride. Crystals grew to full size within 9 days and were cryoprotected in reservoir solution added with 18% (v/v) glycerol and flash-cooled.

The SeMet-labeled Cas12i1 (wild type) binary complex assembled in vivo was crystallized by the hanging-drop vapor diffusion method at 16 °C. Crystals were obtained by mixing 1 μl of complex solution ($A_{280\,nm}$ = 12.3) and 1 μl of reservoir solution (0.1 M sodium citrate (pH 5.2), 17% (w/v) Polyethylene glycol 3350, 2% Tacsimate (pH 4.0), and 0.15 M sodium citrate tribasic dihydrate). Crystals grew to full size within 10 days and were harvested, cryoprotected in reservoir solution containing 16% (v/v) glycerol and flash-cooled.

**Data collection and structure determination**. X-ray diffraction datasets were collected on beamline BL-17U1 of Shanghai Synchrotron Radiation Facility (SSRF) by using the program Blu-Ice and the X-ray wavelength 0.9792 Å. Datasets were automatically processed by the program XDS and XIA2 integrated into data collection platform of the beamline[47,48], and scaled by the program AIMLESS in CCP4[49,50]. The crystal structure of the SeMet-labeled Cas12i1 (D647A mutant) ternary complexes was solved by single-wavelength anomalous dispersion (SAD) method using the program AutoSol in PHENIX[51]. Model building was completed in COOT[52]. Model refinement was performed with REFMAC in CCP4 format[53]. Based on the SeMet-labeled model of the ternary complex, crystal structures of the unlabeled Cas12i1 (D647A mutant or wild type) ternary complex and the wild-type Cas12i1 binary complex were solved by the molecular replacement (MR) method using the program Phaser-MR in PHENIX[51]. Asymmetric units in four kinds of crystals all contain single copy of the ternary or binary complex. Figures of structures were prepared with PyMOL (http://pymol.org). Detailed data collection and refinement statistics were summarized in Supplementary Table 1.

**Site-directed mutagenesis**. Vector encoding the wild-type Cas12i1 was used as a template and oligonucleotides containing the desired mutations were designed (Supplementary Table 4). PCR reactions were carried out to accomplish the site-directed mutagenesis and results were verified by DNA sequencing.

**Plasmid cleavage assay**. Synthetic oligonucleotides (Supplementary Table 5) encoding two strands of target DNA duplex were annealed and inserted between EcoRI and HindIII sites of the pUC-19 vector. Plasmids were verified by DNA sequencing and further linearized by ScaI. Reaction mixtures were prepared by incubating Cas12i1 (final concentration of 0.33 μM) with crRNA (final concentration of 0.33 μM) to form the binary complex at 25 °C for 30 min in assay buffer (50 mM Tris-HCl (pH 8.0), 150 mM NaCl, 10 mM $MgCl_2$, 1 mM DTT). The cleavage assays were then carried out by adding 1000 ng pUC-19 target plasmids at 45 °C for 30 min in reaction mixtures with total volume adjusted to 30 μl. Each cleavage assay was repeated three times independently to confirm the repeatability of the results. Reactions were terminated by adding EDTA and Proteinase K to final concentrations of 100 mM and 0.8 mg/ml at 37 °C for 30 min, respectively. Reaction samples were analyzed by running 1% agarose gels stained with Gel Stain (Transgene Biotech).

**Oligonucleotide cleavage assay**. Target and non-target DNA strands carrying 5′-6FAM and 3′-6FAM fluorescent labels were purchased from Sangon Biotech (Supplementary Table 5) and annealed to form a 70 μM stock. To prepare the cleavage assays, Cas12i1 and crRNA (both with a final concentration of 10 μM) were incubated to form the binary complex at 25 °C for 30 min in assay buffer (50 mM Tris-HCl (pH 8.0), 150 mM NaCl, 2 mM $MgCl_2$, 1 mM DTT). The cleavage assays were performed at 37 °C for 30 min by adding fluorescently labeled double-stranded substrates (final concentration of 5 μM) in reaction mixtures with total volume adjusted to 20 μl. The cleavage assays were repeated independently three times. Reactions were terminated by adding EDTA and Proteinase K to final concentrations of 100 mM and 0.8 mg/ml at 37 °C for 30 min, respectively. Reaction samples were run on a 20% PAGE TBE-urea denaturing gel and the cleavage results were visualized using BIO-RAD Universal Hood II System.

**Cas12i1 activation and *trans*-cleavage assay**. Double-stranded DNA (dsDNA) activators containing 11–16-bp sequences after the PAM duplex and 11–16-nt single-stranded DNA (ssDNA) activators were tested for their ability to trigger the Cas12i1 DNase activity (Supplementary Table 5). A non-specific ssDNA carrying a 5′-6FAM fluorescent label was used as the substrate for the *trans*-cleavage assay (Supplementary Table 5). The Cas12i1 binary complex used in the assay was prepared as described in the protocol for oligonucleotide cleavage assay. The *trans*-cleavage assays were carried out by adding dsDNA or ssDNA activator with a final concentration of 5 μM and fluorescently labeled non-specific ssDNA with a final concentration of 5 μM. The assays were performed at 37 °C for 60 min in reaction mixtures with total volume adjusted to 20 μl. All cleavage assays were repeated independently three times. Reactions were terminated by adding EDTA and Proteinase K to final concentrations of 100 mM and 0.8 mg/ml at 37 °C for 30 min, respectively. Reaction samples were run on a 20% PAGE TBE-urea denaturing gel and results were visualized as using BIO-RAD Universal Hood II System.

**Reporting summary**. Further information on research design is available in the Nature Research Reporting Summary linked to this article.

## Data availability
The atomic coordinates and structure factors of the Cas12i1 pre-cleavage R-loop complex, post-cleavage R-loop complex, and binary complex have been deposited in the Protein Data Bank under the accession codes 7D2L, 7D3J, and 7D8C. Source data are provided with this paper. Other data are available from the corresponding author upon reasonable request.

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

## Acknowledgements

This work was supported by the National Nature Science Foundation of China grants (31770948), the Special Open Fund of Key Laboratory of Experimental Marine Biology, Chinese Academy of Sciences (SKF2020NO1), Marine Economic Development Special Fund of Fujian Province (FJHJF-L-2020-2), the Fujian Provincial Department of Science and Technology (2020Y4007 and 2021H0004), and the High-level personnel introduction grant of Fujian Normal University (Z0210509). The diffraction data were collected at the beamline BL-17U1 of Shanghai Synchrotron Radiation Facility (SSRF).

## Author contributions

B.Z. and S.O. designed the experiments. S.O. and B.Z. supervised the study. B.Z., D.L., and Y.L. prepared Cas12i1, its mutants, and the Cas12i1 complexes. B.Z., V.P., and Y.Y. prepared the in vitro transcribed RNAs. Y.L., D.L., and B.Z. performed the cleavage assays. B.Z., D.L., and Y.L. performed the crystallization screening and optimized the crystallization condition. B.Z. collected X-ray diffraction data and solved the crystal structure. B.Z. wrote the manuscript and prepared the figures. J.C. and J.L. helped with preparing some figures. S.O. and V.P. revised the manuscript.

## Competing interests

The authors declare no competing interests.
