## [Peer Review File · Nature Communications]

REVIEWER COMMENTS

Reviewer #1 (Remarks to the Author):

Zhang B, Luo D et al. report three crystal structures of complexes of Cas12i1, a recently discovered Type V CRISPR-Cas system with predominantly non-target DNA strand nicking activity. The distinct DNA nicking activity of Cas12i and its structural divergence from other Cas12 proteins warrants its structural characterization. Structural and functional characterizations of Cas12i could reveal additional novel properties of this CRISPR-Cas system that could be beneficial in genome editing and nucleic acid detection. For this reason, the research is of interest to a broad audience. The crystallography appears to be technically sound with the note that the resolution of the binary, Cas12i1-crRNA structure is low.

A strength of this manuscript is the comprehensive analysis of and comparison among the crystal structures. Additionally, the authors have generated site-directed mutants of Cas12 and biochemically characterized these mutants to validate specific mechanisms for DNA recognition and cleavage deduced from the crystal structures. The key weakness of the manuscript is that the Discussion section does not compare the structural results obtained here with recent crystal structures of Cas12i reported in Huang X, Sun W. et al (2020 Nature Communications) or the recent cryo-EM structures reported in Zhang H, Zhuang L et al. (2020 NSMB). This discussion is needed to know to what extent the structures and biochemical assays reported in this manuscript reveal new insights into Cas12i structure, dynamics and function versus confirm previously observed mechanisms.

Some additional minor weaknesses and points of clarification are noted. The Abstract refers to a post-cleavage structure but gives the reader too little information on what to expect in this structure. Could the post-cleavage structure also be considered post-cleavage for the non-target DNA strand but also poised for collateral DNA cleavage given that ssDNA is bound in the RuvC active site? Additionally, figures 3a, 3c, 3e, 3f, 4f, 4g, 6f and 6g could all benefit from less detail in the background of the image to allow the reader to focus on the important details in the foreground. Figure 6c should include a scale bar for the length of the vectors so that readers can judge the magnitude of the conformational changes being shown. Line 291 should be re-worded for clarity. A suggestion is 'Two features of the resting state of Cas12i1-crRNA that poise the complex for target duplex binding are the open PAM-interacting cleft and the pre-organized seed region.' The refinement statistics for the binary complex in Supplementary Figure 1 are off by one line.

Reviewer #2 (Remarks to the Author):

Zhang, Luo & Li et al. present three high resolution structures: two of the Cas12i1 R-loop complex (one before and one after DNA cleavage) and one of the Cas12i1-guide RNA (crRNA) complex. The obtained structures in combination with activity assays and mutational analysis allow authors to propose detailed molecular mechanisms of PAM sequence recognition, double stranded DNA (dsDNA) unwinding (R-loop formation), RuvC nuclease activation, and sequential cleavage of non-target and target strands.

Overall, this is an important study that explain not only the molecular mechanism of Cas12i1 mediated dsDNA cleavage but also provides significant insights into the DNA cleavage mechanism by RuvC-containing Cas proteins. The data is technically sound and provides strong evidence for the conclusions presented in this study. This work is of interest to many, because the proposed molecular mechanisms have implications for developing more reliable genome-editing and nucleic acid detection tools.

Although I am very enthusiastic about this work, I have some concerns that I hope the authors will be

able to address in a revised manuscript.

- 1) The domain annotation in this study (Fig. 1 a) differs from the two previously published (PMID: 32895556; 33067443): instead of WED-I the authors call it OBD-I, instead of WED-II they call it OBD-II; TSL-1 and TSL-2 also appear only in this study. Is there any particular reason for this change in naming the domains? Why did authors choose to follow the domain annotation of CasX (PMID: 30718774) instead of Cas12i1 and Cas12i2? I think this brings in unnecessary confusion when one tries to compare the structures and mechanisms of Cas12i orthologs. I would suggest keeping the domain annotations consistent with the previously published studies.
- 2) It is not clear which organism the Cas12i1 characterized here comes from. Is it the same protein as in previously published Zhang et al. study (PMID: 32895556)?
- 3) In the Results sections: Overall architectures of two ternary complexes of Cas12i1, Distinct architecture of the crRNA and its complex with target DNA, Recognition of the crRNA repeat region and Mechanism of the 5'-TTN-3' PAM recognition, besides new observations a substantial part presented here are the same as in previously published study (PMID 32895556). It would be helpful if the authors emphasized which observations are new and how exactly their structures provide more detailed understanding of crRNA and PAM recognition by Cas12i1.
- 4) Based on the structural analysis authors identify several residues (R503, K494, D507, Y509, W511, G584) important for crRNA 5'-end recognition. Do authors know if these residues are important for pre-crRNA processing? Do mutations of these residues affect association of pre-crRNA or dissociation rate of processed crRNA?
- 5) Could authors compare and comment on how similar is the crRNA 5'-end recognition by Cas12i1 and Cas12i2 (PMID: 33067443)?
- 6) When describing Cas12i1-PAM interactions the authors notice that H170 forms H-bonds with dT(-3*) while in a previously published study (PMID: 32895556) it was shown that H170 interacts with dA(-2). Do authors see the same interaction in their structures?
- 7) Line 87: I would suggest referencing supplementary figure 7. "...Cas12a, Cas12b 87 and Cas12e (Fig. 1a-e)." -> "...Cas12a, Cas12b 87 and Cas12e (Fig. 1a-e; Supplementary Fig. 7a-d)."
- 8) Lines 101 through 105: authors compare the overall structure of Cas12i1 to the structures of Cas12b, Cas12a and Cas12e. I would suggest including references to appropriate supplementary figures to demonstrate the similarities and differences.
- 9) Line 128: H518 should be H528.
- 10) The residues L298 and N481 described to be important for PAM recognition are not included in the scheme in Supplementary Fig. 4.
- 11) Lines 179-182: The authors conclude that PAM recognition triggers the zipper motif to obstruct base pairing of TD and NTD to facilitate TD-crRNA base pairing. What data do authors base this conclusion on? Do they see conformational changes of the zipper motif when they compare binary and tertiary structures? If that's the case, an additional figure demonstrating the connection between PAM recognition and the zipper motif activation is necessary.
- 12) Lines 201-203: The authors describe how crRNA-TD heteroduplex is flanked by Cas12i1 on both sides. It would be helpful if authors included a figure emphasizing the flanking they are describing (especially the helix alpha9).
- 13) The authors claim that the crRNA and TD form a 19 nt long heteroduplex. They are able to see the base pairing of dC(20) and dG(20*). However, in the previous studies of Cas12i (PMID: 32895556; 33067443) it was shown that the heteroduplex can be longer than 19 nt. Could authors comment on these different findings? Could it be that the length of the heteroduplex depends on the sequence e.g. A-T base pair at the 20th position would be weaker than G-C and would possibly allow crRNA-DNA hybridization beyond the 19th position?
- 14) The post-cleavage structure is really exciting, especially since the NTD strand is captured by neighboring Cas12i1 and represents ssDNA cleavage in trans state. It is surprising to me that in this structure metal ions are absent from the active site. Could authors comment on that? Did they try to soak these crystals with magnesium containing buffer to see if NTD would be now cleaved in trans?
- 15) Line 259: "... the mechanism of the TD stand..." should be "... the mechanism of the TD strand..."
- 16) Lines 265-257 and Fig. 5b: Could the authors comment on how they can distinguish the trans vs

cis cleavage in this experiment? Also, it seems that the replacement of residues L706-E750 with residues GGSG affect the position of the cleavage. Could authors comment on that and include this observation in the main text?

17) Lines 302-303: The authors claim that the HL motif of the RuvC domain undergoes loop-to-helix transformation. Could authors emphasize this transition in the figure they reference? As is now, it is not clear where this transition happens.

18) While I really enjoyed the comparison of the Cas12i1 molecular mechanism to those of Cas12a, Cas12b and Cas12e, the most important comparison to Cas12i2's mechanism is missing. The study would benefit from more detailed comparison of Cas12i1 and Cas12i2 (PMID: 33067443).

Reviewer #3 (Remarks to the Author):

Dear editor, dear authors,

In their manuscript, Zhang et al. describe the target cleavage mechanism of Cas12i1 based on crystal structures. In this study, binary structures and structures pre-and post cleavage of target DNA are presented, and structural observations are corroborated by biochemical experiments. The analysis is thorough and described in a lot of detail. The manuscript is well written and the insights are interesting.

Major comments

-The study appears largely confirmatory of the previous Cas12i1 cryo-EM study performed by Zhang et al, (2020; NSMB), although the current study provides slightly higher resolution (but comparable to the study of Cas12i2 by Hueng et al (2020; Nat Com. Cas12i2)).

However, the authors state "More recently, cryo-EM and crystal structures of Cas12i in crRNA-bound and crRNA-target-bound states have been reported^{27,28}. However, a number of key questions concerning the target DNA duplex unwinding, R-loop formation, target DNA duplex cleavage, key residues responsible for the PAM recognition in Cas12i1 and the mechanism of the DNase activation remain unanswered or require further investigation." After that, comparison with the other structures occurs only 3 times in the entire manuscript. While I do think that the authors do reveal novel features of Cas12i that are previously not described in the other structural analyses, it is difficult to distil the novel insights from all the other insights which have also been revealed in the earlier studies. The study lacks thorough comparison with these earlier determined structures.

I suggest to revise the manuscript in order to make the novel insights obtained from the thorough structural analysis and biochemical experiments more clear for the reader. In addition, structure-structure comparisons should be made to indicate what is similar in the different Cas12i structures, and what is not.

-G235 and A236 play a role in PAM determination by excluding GC base pairs at this position. The authors have mutated G235 and A236 to Ala and Lys, respectively. This results in loss of target cleavage, and the authors implicate these residues in PAM recognition. However, I would suggest to change the phrasing, as their experiment demonstrates PAM and thereby DNA exclusion, rather than loss of specific recognition. Also, have the authors attempted to model TA (instead of AT) base pairs at the same position? Would they also be excluded like GC base pairs? Is this type of PAM recognition also observed in the other Cas12i structures?

Minor comments

-The gap between Rwork and Rfree reported in Supp. Table 1 is too large for the pre-cleavage R-loop complex and it suggests over-refining (19.1/25.2).

-I find it rather confusing that domains have been renamed compared to the previous manuscripts in which Cas12i1 and Cas12i2 are structurally characterized. I highly suggest to keep the same nomenclature for things that are the same, to make comparison easier.

Point-by-point response to Reviewers' comments

The authors thank the reviewers for professional and insightful review. Point-by-point responses and changes in the revised manuscript are highlighted in blue color, and the deleted sentences in the revised manuscript are highlighted in deep orange color. Following the reviewers' suggestion, the authors have added a new subsection of "Structural analysis of Cas12i1 and Cas12i2 complexes" in the Discussion section of the revised manuscript. Besides, the authors have renamed the OBD, Helical, TSL-1 and TSL-2 domains to the WED, Helical-I, Helical-II and Nuc domains in the revised manuscript, respectively. The authors reorganized the first four subsections in the Results section of the manuscript into three subsections and removed the repeated information related to the previous study (Nature Structural & Molecular Biology, 2020, PMID: 32895556). Other changes in the manuscript have been described in the following point-by-point responses.

REVIEWER COMMENTS

Reviewer #1 (Remarks to the Author):

Zhang B, Luo D et al. report three crystal structures of complexes of Cas12i1, a recently discovered Type V CRISPR-Cas system with predominantly non-target DNA strand nicking activity. The distinct DNA nicking activity of Cas12i and its structural divergence from other Cas12 proteins warrants its structural characterization. Structural and functional characterizations of Cas12i could reveal additional novel properties of this CRISPR-Cas system that could be beneficial in genome editing and nucleic acid detection. For this reason, the research is of interest to a broad audience. The crystallography appears to be technically sound with the note that the resolution of the binary, Cas12i1-crRNA structure is low.

Point-by-point responses to Reviewer-1's comments:

A strength of this manuscript is the comprehensive analysis of and comparison among the crystal structures. Additionally, the authors have generated site-directed mutants of Cas12 and biochemically characterized these mutants to validate specific mechanisms for DNA recognition and cleavage deduced from the crystal structures. The key weakness of the manuscript is that the Discussion section does not compare the structural results obtained here with recent crystal structures of Cas12i reported in Huang X, Sun W. et al (2020 Nature Communications) or the recent cryo-EM structures reported in Zhang H, Zhuang L et al. (2020 NSMB). This discussion is needed to know to what extent the structures and biochemical assays reported in this manuscript reveal new insights into Cas12i structure, dynamics and function versus confirm previously observed mechanisms.

Authors' response:

Following the reviewer's suggestion, the authors have added a new subsection of "Structural analysis of Cas12i1 and Cas12i2 complexes" in the Discussion section of the revised manuscript.

The authors started to write this manuscript at the beginning of August 2020. The two structural studies on Cas12i1 and Cas12i2 mentioned by the reviewer were released online on September and October of 2020, respectively. Fortunately, the three crystal structures of Cas12i1 reported in our study provided novelty and additional information. To avoid direct criticism of findings of the two previous studies and shorten the length of this manuscript, the authors did not make an in-depth comparison of the structural results of the present and two previous studies. However, the authors have discussed the novel findings of this study and compared results of this study with two previous studies in the cover letter to the editor. To make the major findings of this study clearer for the reviewer, the authors provided the revised content from the former cover letter, and the main points mentioned below were also been included into the subsection of “Structural analysis of Cas12i1 and Cas12i2 complexes” in the Discussion section of the revised manuscript.

The major findings in our study are:

1. We solved the crystal structures of the Cas12i1 R-loop ternary complex before and after target dsDNA cleavage at 2.75 Å and 2.45 Å resolution, respectively. In both R-loop complexes, the target DNA (TD) strand of the target DNA duplex is unwound from the non-target DNA (NTD) strand and forms a 19-bp heteroduplex with the crRNA guide region. In the pre-cleavage R-loop complex, the nucleotide dG(20*) of the NTD strand swings to interact with dC(20) of the TD strand. Possibly due to the incomplete state of target dsDNA nucleotides in their ternary complexes and the longer crRNA guide region used to prepare the Cas12i1 and Cas12i2 ternary complexes, the 28-bp and 26-bp heteroduplexes have been reported to be accommodated in the previously reported Cas12i1 and Cas12i2 ternary complexes, respectively (Heng Zhang et al., *Nature Structural & Molecular Biology*, 2020; Xue Huang et al., *Nature Communications*, 2020). This implies that the complete R-loop structure in Cas12i will affect the reasonable length of the heteroduplex formed by the TD strand and the crRNA guide region.
2. A key motif, which we term the zipper motif (residues 160–177aa of the Helical-I domain), and two groups of “phosphate lock” interactions (residues R12 and K483, R535 and K923) have been identified to facilitate the target DNA duplex unwinding.
3. The previously unreported residues G235 and A236 of the PI domain play critical roles in PAM determination.
4. Cas12i1 cleaves the NTD strand prior to the TD strand. The NTD strand was found to be successively cleaved 13–15 nucleotides after the PAM duplex, whereas the TD strand was cleaved 24 nucleotides after the PAM duplex. This information concerning target dsDNA cleavage was not reported by Heng Zhang et al. (*Nature Structural & Molecular Biology*, 2020). An unusual result reported by Xue Huang et al. (*Nature Communications*, 2020) indicates that the NTD strand is primarily cleaved 31 nucleotides after the PAM duplex by Cas12i2. Such cleavage position on the NTD strand in Cas12i2 was not observed in Cas12i1, Cas12a, Cas12b and Cas12e.
5. In our post-cleavage R-loop complex, the NTD and TD strands are already cleaved by Cas12i1. The PAM-distal region of the cleavage product is released and the PAM-proximal region of target DNA duplex is retained in Cas12i1. A segment of the retained NTD strand stretches out of Cas12i1 and is captured by the neighboring ternary complex, thus providing additional information regarding the pre-cleavage state of the Cas12i1 trans cleavage.

6. In the pre-cleavage R-loop complex, a loop region (residues 724–737aa) in the Helical-II domain helps the 5'-terminal of the TD strand to pair again with the 3'-terminal of the NTD strand. This loop region also facilitates the loading of TD strand beyond the heteroduplex into the Cas12i1 DNase active site.
7. In our crystal structure of the Cas12i1 binary complex, the crRNA seed region was found to interact with a complementary 3-nt pseudo target originating from *E. coli* cells in which the binary complex was expressed. These base pairings within the seed region mimic a scene of target interrogation in Cas12i1.
8. The overlay of the Cas12i1 binary and ternary complex reveals that the PI domain, the Helical-I domain and the helix-loop (HL) motif (residues 895–915aa) of the RuvC domain go through remarkable conformational rearrangement during transition to ternary complex. Upon the target DNA duplex binding, the PAM-interacting cleft undergoes an “open to closed” conformational transformation. Besides, the PAM-distal α -helix bundle of the Helical-I domain and the HL motif are rearranged so that they no longer obstruct access to the RuvC active site, thereby unleashing the Cas12i1 DNase activity. Heng Zhang et al. (Nature Structural & Molecular Biology, 2020) only reported the conformational rearrangement of the HL motif in Cas12i1. Xue Huang et al. (Nature Communications, 2020) reported the conformational rearrangement of the Helical-II domain in Cas12i2 (the counterpart of the PAM-distal α -helix bundle of the Helical-I domain in Cas12i1). Due to lack of the PI domain in their binary complexes, the conformational movement of the PI domain was not observable in these studies.
9. We show that substrate cleavage by Cas12i1 is temperature and metal-ion dependent.
10. We discussed the structural divergence of the CRISPR-Cas12 family members including Cas12a, Cas12b, Cas12e and Cas12i. In addition, the critical motifs facilitating the R-loop formation in Cas12 effectors were compared and analyzed. Based on our structural and biochemical studies, we propose a multi-step model to explain the process of catalytic activation of Cas12i1.

Some additional minor weaknesses and points of clarification are noted.

The Abstract refers to a post-cleavage structure but gives the reader too little information on what to expect in this structure.

Authors' response:

Following the reviewer's suggestion, the authors have revised the relevant sentences in the Abstract, which now state the following: “Here we report the crystal structures of the Cas12i1 R-loop complexes before and after target DNA cleavage to elucidate the mechanisms underlying target DNA duplex unwinding, R-loop formation and cis cleavage. The structure of the R-loop complex after target DNA cleavage also provides information regarding trans cleavage.”

Could the post-cleavage structure also be considered post-cleavage for the non-target DNA strand but also poised for collateral DNA cleavage given that ssDNA is bound in the RuvC active site?

Authors' response:

Yes, the post-cleavage R-loop complex not only represents a state of the R-loop complex after the cis cleavage of the target duplex, but also provides additional information about the pre-cleavage state for the trans cleavage or collateral cleavage.

Additionally, figures 3a, 3c, 3e, 3f, 4f, 4g, 6f and 6g could all benefit from less detail in the background of the image to allow the reader to focus on the important details in the foreground. Figure 6c should include a scale bar for the length of the vectors so that readers can judge the magnitude of the conformational changes being shown.

Authors' response:

The authors thank the reviewer for this suggestion. Following the reviewer's suggestion, the authors have revised Figure 2c, 2d, 3a, 3c, 3e, 3f, 4f, 4g, 6f and 6g in order to reduce the background. Also, the authors have added a scale bar for the length of the vectors in Figure 6c.

Line 291 should be re-worded for clarity. A suggestion is 'Two features of the resting state of Cas12i1-crRNA that poise the complex for target duplex binding are the open PAM-interacting cleft and the pre-organized seed region.'

Authors' response:

Following the reviewer's suggestion, the authors have revised this sentence for clarity as the following: "Two key features of the resting state of the Cas12i1-crRNA binary complex that poise it for the target duplex binding are the open PAM-interacting cleft and preorganized seed region."

The refinement statistics for the binary complex in Supplementary Figure 1 are off by one line.

Authors' response:

The authors thank the reviewer for reminding us about this point and we have revised it.

Reviewer #2 (Remarks to the Author):

Zhang, Luo & Li et al. present three high resolution structures: two of the Cas12i1 R-loop complex (one before and one after DNA cleavage) and one of the Cas12i1-guide RNA (crRNA) complex. The obtained structures in combination with activity assays and mutational analysis allow authors to propose detailed molecular mechanisms of PAM sequence recognition, double stranded DNA (dsDNA) unwinding (R-loop formation), RuvC nuclease activation, and sequential cleavage of non-target and target strands.

Overall, this is an important study that explain not only the molecular mechanism of Cas12i1 mediated dsDNA cleavage but also provides significant insights into the DNA cleavage mechanism by RuvC-containing Cas proteins. The data is technically sound and provides strong evidence for the conclusions presented in this study. This work is of interest to many, because the proposed molecular mechanisms have implications for developing more reliable genome-editing and nucleic acid detection tools.

Although I am very enthusiastic about this work, I have some concerns that I hope the authors will be able to address in a revised manuscript.

- 1) The domain annotation in this study (Fig. 1 a) differs from the two previously published (PMID: 32895556; 33067443): instead of WED-I the authors call it OBD-I, instead of WED-II they call it OBD-II; TSL-1 and TSL-2 also appear only in this study. Is there any particular reason for this change in naming the domains? Why did authors choose to follow the domain annotation of CasX (PMID: 30718774) instead of Cas12i1 and Cas12i2? I think this brings in unnecessary confusion when one tries to compare the structures and mechanisms of Cas12i orthologs. I would suggest keeping the domain annotations consistent with the previously published studies.

Authors' response:

Following the reviewer's suggestion, the authors renamed the OBD, Helical, TSL-1 and TSL-2 domains in the former manuscript to the WED, Helical-I, Helical-II and Nuc domains in the revised manuscript, respectively.

The previously named "Nuc" domain in Cas12a was supposed to function as an endonuclease domain, whereas this domain actually did not have the DNase activity. That was the reason that Prof. Jennifer A. Doudna's group suggested to rename the Nuc domain as the target-strand loading (TSL) domain for such domain annotation in Cas12 family and they used the annotation of the TSL domain in their study of the CasX complexes. Therefore, the authors followed this suggestion while preparing the previous version of this manuscript.

The REC2 domain of Cas12i1 in the study reported by Heng Zhang et al. (Nature Structural & Molecular Biology, 2020, PMID: 32895556) should be included into the NUC lobe, but not the REC lobe, and it would be plausible to rename the REC2 domain to the Helical-II domain. Thus, the authors use the domain annotation of the Helical-I and Helical-II domains to replace the name of the REC1 and REC2 domains in the revised manuscript.

- 2) It is not clear which organism the Cas12i1 characterized here comes from. Is it the same protein as in previously published Zhang et al. study (PMID: 32895556)?

Authors' response:

The authors could not find the organism from which the Cas12i1 originates in the original paper reporting Cas12i1 and Cas12i2 by Winston X. Yan et al. (Science, 2019, PMID: 30523077). Search result using Cas12i1 as query sequence on NCBI BLAST indicates that Cas12i1 originates from *Lachnospiraceae bacterium* ND2006.

Cas12i1 in this study is the same protein as in the previous study by Heng Zhang et al. (Nature Structural & Molecular Biology, 2020, PMID: 32895556).

- 3) In the Results sections: Overall architectures of two ternary complexes of Cas12i1, Distinct architecture of the crRNA and its complex with target DNA, Recognition of the crRNA repeat region and Mechanism of the 5'-TTN-3' PAM recognition, besides new observations a substantial part presented here are the same as in previously published study (PMID 32895556). It would be helpful if the authors emphasized which observations are new and how exactly their structures provide more detailed understanding of crRNA and PAM recognition by Cas12i1.

Authors' response:

The authors thank the reviewer for this suggestion. We have reorganized the first four subsections in the Results section of the manuscript into three subsections and removed the repeated information related to the previous study (Nature Structural & Molecular Biology, 2020, PMID: 32895556) as much as possible, and also kept the new observations and analysis of the present study.

- 4) Based on the structural analysis authors identify several residues (R503, K494, D507, Y509, W511, G584) important for crRNA 5'-end recognition. Do authors know if these residues are important for pre-crRNA processing? Do mutations of these residues affect association of pre-crRNA or dissociation rate of processed crRNA?

Authors' response:

To answer these two questions, the authors prepared the Cas12i1 K494A, R503A, D507A, Y509A, W511A and G584A mutants. The pre-crRNA containing a repeat-spacer-repeat sequence was designed and purified.

To check if these mutants are important for the pre-crRNA processing, the authors carried out the pre-crRNA cleavage assays. Reaction mixtures were prepared by incubating 6.0 μ M purified wild-type Cas12i1 or its mutants with 3.0 μ M pre-crRNA in the assay buffer (25 mM Tris-HCl (pH 7.5), 150 mM NaCl, 2 mM MgCl₂, 1 mM DTT). Total volume of the reaction mixture was adjusted to 20 μ l. The cleavage was allowed to proceed at 37°C for 30 minutes, after which the reactions were terminated by adding EDTA and Proteinase K to final concentrations of 100 mM and 0.8 mg/ml at 37°C for 30 minutes. Samples were separated by running on a 20% PAGE TBE-urea denaturing gel and the cleavage products were visualized with Gel Stain. The results show that the D507A and W511A mutations have decreased the pre-crRNA processing capability.

To investigate the association capacities between pre-crRNA and the wild-type Cas12i1 or its mutants, the authors carried out the EMSA assays. The authors incubated the wild-type Cas12i1 or its mutants with pre-crRNA at 22°C for 30 minutes in the assay buffer (25 mM Tris-HCl (pH 7.5), 150 mM NaCl, 2 mM MgCl₂, 1 mM DTT). Total volume of the assay mixture was adjusted to 20 µl. The final concentration of pre-crRNA was 3.0 µM, and three final concentrations of wild-type Cas12i1 or its mutants were 1.5 µM, 3.0 µM and 6.0 µM. Samples were separated by running on a 6% native PAGE gel and 0.5xTBE buffer were used for electrophoresis. The native PAGE gels were visualized with Gel Stain. The results indicate that the wild-type Cas12i1 and its mutants have similar and comparable association capacity with pre-crRNA. Therefore, the Cas12i1 K494A, R503A, D507A, Y509A, W511A and G584A mutations have little effect to affect the association of Cas12i1 with pre-crRNA.

(a) pre-crRNA cleavage results

(b) EMSA results

Figure | **a**, the pre-crRNA cleavage results. The wild-type Cas12i1 and its mutants were used to perform the pre-crRNA cleavage assays and the results were presented by running a 20% PAGE TBE-urea denaturing gel. **b**, the EMSA results. The wild-type Cas12i1 and its mutants were used to perform the EMSA assays and the results were presented by running a 6% native PAGE gel.

- 5) Could authors compare and comment on how similar is the crRNA 5'-end recognition by Cas12i1 and Cas12i2 (PMID: 33067443)?

Authors' response:

Following the reviewer's suggestion, the authors have prepared the following figure to explore the similarity of the crRNA 5'-end recognition by Cas12i1 and Cas12i2. The overall conformations of the crRNA 5'-end containing the nucleotides A(-23)-U(-18) in the Cas12i1

and Cas12i2 ternary complexes resemble each other to a certain degree. However, the nucleotide sequences of the crRNA 5'-end in Cas12i1 and Cas12i2 systems are obviously different. Besides, details of the recognition pattern and the conformational orientations of the nucleotides (-22)-(-19) are different in Cas12i1 and Cas12i2. There are more interactions between residues and the base rings of the crRNA 5'-end nucleotides in Cas12i2.

Figure | The Cas12i1 pre-cleavage R-loop complex (PDB ID: 7D2L) and the Cas12i2 ternary complex (PDB ID: 6LTR) were used to prepare this figure.

- 6) When describing Cas12i1-PAM interactions the authors notice that H170 forms H-bonds with dT(-3*) while in a previously published study (PMID: 32895556) it was shown that H170 interacts with dA(-2). Do authors see the same interaction in their structures?

Authors' response:

The authors have checked the structural model of the Cas12i1 ternary complex (PDB ID: 6W5C) reported in the previously published study (Nature Structural & Molecular Biology, 2020, PMID: 32895556). The authors found that residue H170 had the similar hydrogen-bonding mode as in the present study (shown in the figure below). Thus, the interaction between H170 and dA(-2) reported in the previously published study (Nature Structural & Molecular Biology, 2020, PMID: 32895556) is likely a mistake.

Figure | Residue H170 forms hydrogen bond with dT(-3*) in the structural model of the Cas12i1 ternary complex (PDB ID: 6W5C) and does not form hydrogen bond with dA(-2) (the distance is 3.55Å between them and colored in green dashed line).

- 7) Line 87: I would suggest referencing supplementary figure 7. "...Cas12a, Cas12b 87 and Cas12e (Fig. 1a-e)." -> "...Cas12a, Cas12b 87 and Cas12e (Fig. 1a-e; Supplementary Fig. 7a-d)."

Authors' response:

Following the reviewer's suggestion, the authors have renumbered the Supplementary Figures and revised this item.

- 8) Lines 101 through 105: authors compare the overall structure of Cas12i1 to the structures of Cas12b, Cas12a and Cas12e. I would suggest including references to appropriate supplementary figures to demonstrate the similarities and differences.

Authors' response:

The authors have renamed the TSL-1 and TSL-2 domains to the Helical-II and Nuc domains in the revised manuscript. In order to reduce the repeated information related to the previous study (Nature Structural & Molecular Biology, 2020, PMID: 32895556), sentences in Lines 101 to 105 have been deleted in the revised manuscript.

- 9) Line 128: H518 should be H528.

Authors' response:

The authors thank the reviewer for pointing out this mistake. We revised it accordingly.

- 10) The residues L298 and N481 described to be important for PAM recognition are not included in the scheme in Supplementary Fig. 4.

Authors' response:

Following the reviewer's suggestion, the authors have added interactions between residues L298 and N481 and the PAM sequence in the Supplementary Fig. 4.

- 11) Lines 179-182: The authors conclude that PAM recognition triggers the zipper motif to obstruct base pairing of TD and NTD to facilitate TD-crRNA base pairing. What data do authors base this conclusion on? Do they see conformational changes of the zipper motif when

they compare binary and tertiary structures? If that's the case, an additional figure demonstrating the connection between PAM recognition and the zipper motif activation is necessary.

Authors' response:

The authors thank the reviewer for the logical analysis and for raising this question. The authors also thought that the meaning of the sentence in Lines 179 to 182 was not logically rigorous enough and the word "trigger" in the sentence needs additional information to support it. The authors revised this sentence to state the following: "Therefore, binding of the target duplex and subsequent recognition of the correct PAM sequence by Cas12i1 could make the PAM-proximal region of the target duplex approach the zipper motif, and the zipper motif could generate a spatial obstruction towards base pairing within the PAM proximal region of the target duplex and contribute to the nucleation between the TD strand and the seed region of the crRNA guide".

During the transition from the binary complex to the pre-cleavage R-loop complex, the conformation of the PI domain undergoes a remarkable rearrangement, and the conformation the zipper motif only undergoes a slight movement.

- 12) Lines 201-203: The authors describe how crRNA-TD heteroduplex is flanked by Cas12i1 on both sides. It would be helpful if authors included a figure emphasizing the flanking they are describing (especially the helix alpha9).

Authors' response:

Following the reviewer's suggestion, the authors have revised Fig. 3 for emphasizing such flanking and helix alpha9 in the revised manuscript.

- 13) The authors claim that the crRNA and TD form a 19 nt long heteroduplex. They are able to see the base pairing of dC(20) and dG(20*). However, in the previous studies of Cas12i (PMID: 32895556; 33067443) it was shown that the heteroduplex can be longer than 19 nt. Could authors comment on these different findings? Could it be that the length of the heteroduplex depends on the sequence e.g. A-T base pair at the 20th position would be weaker than G-C and would possibly allow crRNA-DNA hybridization beyond the 19th position?

Authors' response:

The authors thank the reviewer for this good question. When we saw the lengths of heteroduplexes reported by two previous studies, we also wanted to understand the possible reason behind the longer heteroduplexes (28-bp in Cas12i1 and 26-bp in Cas12i2). We gave our explanations in the new subsection entitled "Structural analysis of Cas12i1 and Cas12i2 complexes" in the Discussion section of the revised manuscript. The authors speculate that the possible main reason is partial fragments of the target duplexes used to prepare the samples of the ternary complexes in two previous studies. In contrast, the authors used the full R-loop fragments to prepare the ternary complexes in the present study. Besides, a possible minor reason was that the longer crRNA guide regions was used *in vitro* to prepare the Cas12i1 and Cas12i2 ternary complexes by two previous studies, whereas the author used the *in vivo* method to acquire the binary complex at first and then prepared the full R-loop ternary

complexes *in vitro*.

Of note, the authors also have additional unpublished crystal-structure data (2.40 Å resolution) to support the 19-bp heteroduplex in the Cas12i1 and the result can answer the question asked by the reviewer as shown in the following figure. Even when the 20th G-C base pair in this study was replaced by two unpaired 20th nucleotides A-C, the 20th nucleotide (cytosine) of the TD strand also swung towards the 20th nucleotide (adenine) of the NTD strand beyond the 19-bp heteroduplex. This implied that the relative complete R-loop structure in Cas12i will affect the reasonable length of the heteroduplex formed by the TD strand and the crRNA guide region.

- 14) The post-cleavage structure is really exciting, especially since the NTD strand is captured by neighboring Cas12i1 and represents ssDNA cleavage in trans state. It is surprising to me that in this structure metal ions are absent from the active site. Could authors comment on that? Did they try to soak these crystals with magnesium containing buffer to see if NTD would be now cleaved in trans?

Authors' response:

The authors have seen that one or two magnesium ions were observed in the RuvC active site of two Cas12i2 ternary complexes (Nature Communications, 2020, PMID: 33067443). We

also wished to solve magnesium ions in the RuvC active site of the post-cleavage R-loop structure of the Cas12i1 ternary complex, but the electron density surrounding the RuvC active site in the structural model did not allow us to do so. Magnesium ion normally formed six coordination bonds, which was also not supported in the present structural model. The authors thought that binding of magnesium ions inside the Cas12i RuvC active site might be a dynamic process with balance of three magnesium ion-binding states: zero, one or two magnesium ions presented in the Cas12i RuvC active site.

The authors did not soak the crystal with a buffer containing high-concentration magnesium ions, but the buffer system containing 2 mM of magnesium ions was used to prepare the post-cleavage R-loop complex of Cas12i1 for the crystallization.

- 15) Line 259: "... the mechanism of the TD stand..." should be "... the mechanism of the TD strand..."

Authors' response:

The authors thank the reviewer for finding this misspelling, which we revised.

- 16) Lines 265-267 and Fig. 5b: Could the authors comment on how they can distinguish the trans vs cis cleavage in this experiment? Also, it seems that the replacement of residues L706-E750 with residues GGSG affect the position of the cleavage. Could authors comment on that and include this observation in the main text?

Authors' response:

The authors distinguished the cis vs trans cleavage based on the cleavage pattern obtained from the assay shown in Figure 4b. For the clarity, we labeled the cis cleavage of the NTD and TD strands in the green and blue triangles, respectively, and the trans cleavage in the red triangles for Figure 5b.

The authors also observed that the $\Delta 724-737$ aa and $\Delta 706-750$ aa mutants altered the cleavage pattern. A possible explanation was that such mutations were not far from the RuvC active site and they might change the loading mode or the loading conformation that the substrate was loaded into the RuvC active site. Following the reviewer's suggestion, the authors included this observation in the revised manuscript.

Fig. 5b

17) Lines 302-303: The authors claim that the HL motif of the RuvC domain undergoes loop-to-helix transformation. Could authors emphasize this transition in the figure they reference? As is now, it is not clear where this transition happens.

Authors' response:

The loop-to-helix transition of the HL motif happens within a short α -helix containing 10 amino-acid residues. Considering that the phenomenon of the loop-to-helix transition was not so obvious, the authors deleted “undergoes loop-to-helix transformation” in the revised manuscript.

18) While I really enjoyed the comparison of the Cas12i1 molecular mechanism to those of Cas12a, Cas12b and Cas12e, the most important comparison to Cas12i2's mechanism is missing. The study would benefit from more detailed comparison of Cas12i1 and Cas12i2 (PMID: 33067443).

Authors' response:

Following the reviewer's suggestion, the authors have added a new subsection entitled “Structural analysis of Cas12i1 and Cas12i2 complexes” to the Discussion section of the revised manuscript.

Reviewer #3 (Remarks to the Author):

Dear editor, dear authors,

In their manuscript, Zhang et al. describe the target cleavage mechanism of Cas12i1 based on crystal structures. In this study, binary structures and structures pre-and post cleavage of target DNA are presented, and structural observations are corroborated by biochemical experiments. The analysis is thorough and described in a lot of detail. The manuscript is well written and the insights are interesting.

Major comments

-The study appears largely confirmatory of the previous Cas12i cryo-EM study performed by Zhang et al, (2020; NSMB), although the current study provides slightly higher resolution (but comparable to the study of Cas12i2 by Huang et al (2020; Nat Com. Cas12i2)).

However, the authors state "More recently, cryo-EM and crystal structures of Cas12i in crRNA-bound and crRNA-target-bound states have been reported^{27,28}. However, a number of key questions concerning the target DNA duplex unwinding, R-loop formation, target DNA duplex cleavage, key residues responsible for the PAM recognition in Cas12i1 and the mechanism of the DNase activation remain unanswered or require further investigation." After that, comparison with the other structures occurs only 3 times in the entire manuscript. While I do think that the authors do reveal novel features of Cas12i that are previously not described in the other structural analyses, it is difficult to distil the novel insights from all the other insights which have also been revealed in the earlier studies. The study lacks thorough comparison with these earlier determined structures.

I suggest to revise the manuscript in order to make the novel insights obtained from the thorough structural analysis and biochemical experiments more clear for the reader. In addition, structure-structure comparisons should be made to indicate what is similar in the different Cas12i structures, and what is not.

Authors' response:

Following the reviewer's suggestion, the authors have added a new subsection of "Structural analysis of Cas12i1 and Cas12i2 complexes" in the Discussion section of the revised manuscript to make a comparison of Cas12i structures between this study and two previous studies. The authors also reorganized the first four subsections in the Results section of the manuscript into three subsections and removed the repeated information related to the previous study (Nature Structural & Molecular Biology, 2020, PMID: 32895556).

The authors started to write this manuscript at the beginning of August 2020. The two structural studies on Cas12i1 and Cas12i2 mentioned by the reviewer were released online on September and October of 2020, respectively. Fortunately, the three crystal structures of Cas12i1 reported in our study provided novelty and additional information. To avoid direct criticism of findings of the two previous studies and shorten the length of this manuscript, the authors did not make an in-depth comparison of the structural results of the present and two previous studies. However, the authors have discussed the novel findings of this study and compared results of this study with two

previous studies in the cover letter to the editor. To make the major findings of this study clearer for the reviewer, the authors provided the revised content from the former cover letter, and the main points mentioned below were also been included into the subsection of “Structural analysis of Cas12i1 and Cas12i2 complexes” in the Discussion section of the revised manuscript.

The major findings in our study are:

1. We solved the crystal structures of the Cas12i1 R-loop ternary complex before and after target dsDNA cleavage at 2.75 Å and 2.45 Å resolution, respectively. In both R-loop complexes, the target DNA (TD) strand of the target DNA duplex is unwound from the non-target DNA (NTD) strand and forms a 19-bp heteroduplex with the crRNA guide region. In the pre-cleavage R-loop complex, the nucleotide dG(20*) of the NTD strand swings to interact with dC(20) of the TD strand. Possibly due to the incomplete state of target dsDNA nucleotides in their ternary complexes and the longer crRNA guide region used to prepare the Cas12i1 and Cas12i2 ternary complexes, the 28-bp and 26-bp heteroduplexes have been reported to be accommodated in the previously reported Cas12i1 and Cas12i2 ternary complexes, respectively (Heng Zhang et al., *Nature Structural & Molecular Biology*, 2020; Xue Huang et al., *Nature Communications*, 2020). This implies that the complete R-loop structure in Cas12i will affect the reasonable length of the heteroduplex formed by the TD strand and the crRNA guide region.
2. A key motif, which we term the zipper motif (residues 160–177aa of the Helical-I domain), and two groups of “phosphate lock” interactions (residues R12 and K483, R535 and K923) have been identified to facilitate the target DNA duplex unwinding.
3. The previously unreported residues G235 and A236 of the PI domain play critical roles in PAM determination.
4. Cas12i1 cleaves the NTD strand prior to the TD strand. The NTD strand was found to be successively cleaved 13–15 nucleotides after the PAM duplex, whereas the TD strand was cleaved 24 nucleotides after the PAM duplex. This information concerning target dsDNA cleavage was not reported by Heng Zhang et al. (*Nature Structural & Molecular Biology*, 2020). An unusual result reported by Xue Huang et al. (*Nature Communications*, 2020) indicates that the NTD strand is primarily cleaved 31 nucleotides after the PAM duplex by Cas12i2. Such cleavage position on the NTD strand in Cas12i2 was not observed in Cas12i1, Cas12a, Cas12b and Cas12e.
5. In our post-cleavage R-loop complex, the NTD and TD strands are already cleaved by Cas12i1. The PAM-distal region of the cleavage product is released and the PAM-proximal region of target DNA duplex is retained in Cas12i1. A segment of the retained NTD strand stretches out of Cas12i1 and is captured by the neighboring ternary complex, thus providing additional information regarding the pre-cleavage state of the Cas12i1 trans cleavage.
6. In the pre-cleavage R-loop complex, a loop region (residues 724–737aa) in the Helical-II domain helps the 5'-terminal of the TD strand to pair again with the 3'-terminal of the NTD strand. This loop region also facilitates the loading of TD strand beyond the heteroduplex into the Cas12i1 DNase active site.
7. In our crystal structure of the Cas12i1 binary complex, the crRNA seed region was found to interact with a complementary 3-nt pseudo target originating from *E. coli* cells in which the binary complex was expressed. These base pairings within the seed region mimic a scene of target interrogation in Cas12i1.

8. The overlay of the Cas12i1 binary and ternary complex reveals that the PI domain, the Helical-I domain and the helix-loop (HL) motif (residues 895–915aa) of the RuvC domain go through remarkable conformational rearrangement during transition to ternary complex. Upon the target DNA duplex binding, the PAM-interacting cleft undergoes an “open to closed” conformational transformation. Besides, the PAM-distal α -helix bundle of the Helical-I domain and the HL motif are rearranged so that they no longer obstruct access to the RuvC active site, thereby unleashing the Cas12i1 DNase activity. Heng Zhang et al. (Nature Structural & Molecular Biology, 2020) only reported the conformational rearrangement of the HL motif in Cas12i1. Xue Huang et al. (Nature Communications, 2020) reported the conformational rearrangement of the Helical-II domain in Cas12i2 (the counterpart of the PAM-distal α -helix bundle of the Helical-I domain in Cas12i1). Due to lack of the PI domain in their binary complexes, the conformational movement of the PI domain was not observable in these studies.
9. We show that substrate cleavage by Cas12i1 is temperature and metal-ion dependent.
10. We discussed the structural divergence of the CRISPR-Cas12 family members including Cas12a, Cas12b, Cas12e and Cas12i. In addition, the critical motifs facilitating the R-loop formation in Cas12 effectors were compared and analyzed. Based on our structural and biochemical studies, we propose a multi-step model to explain the process of catalytic activation of Cas12i1.

-G235 and A236 play a role in PAM determination by excluding GC base pairs at this position. The authors have mutated G235 and A236 to Ala and Leu, respectively. This results in loss of target cleavage, and the authors implicate these residues in PAM recognition. However, I would suggest to change the phrasing, as their experiment demonstrates PAM and thereby DNA exclusion, rather than loss of specific recognition. Also, have the authors attempted to model TA (instead of AT) base pairs at the same position? Would they also be excluded like GC base pairs? Is this type of PAM recognition also observed in the other Cas12i structures?

Authors' response:

The authors thank the reviewer for the professional suggestion. Following the reviewer's suggestion, the authors have revised the term “PAM recognition” to “PAM determination” in the revised manuscript. Also, the authors modeled two TA base pairs (instead of two AT base pairs) in the PAM sequence of Cas12i1. Both modeled TA base pairs have the steric clashes with residues of Cas12i1 as shown in the following figure.

Figure | **a**, The modeled dT(-3):dA(-3*) base pair in Cas12i1 would generate steric clashes with the side chain of residue N481. **b**, The modeled dT(-2):dA(-2*) base pair in Cas12i1 would generate steric clashes with the side chain of residue S482.

In Cas12i2, two modeled GC base pairs in the PAM sequence would also clash with residues G231 and A232 (the counterparts of G235 and A236 in Cas12i1) as shown in the following figure, whereas two modeled TA base pairs (instead of two AT base pairs) in the PAM sequence of Cas12i2 will not clash with the surrounding residues in Cas12i2.

Figure | **a**, The modeled dG(-3):dC(-3*) base pair in Cas12i2 (PDB ID: 6LTR) would generate a steric clash with the backbone nitrogen atom of residue A232. **b**, The modeled dG(-2):dC(-2*) base pair in Cas12i2 would generate a steric clash with the side chain of residue A232.

Minor comments

-The gap between R_{work} and R_{free} reported in Supp. Table 1 is too large for the pre-cleavage R-loop complex and it suggests over-refining (19.1/25.2).

Authors' response:

Following the reviewer's suggestion, the authors revised the structural model of the pre-cleavage R-loop complex to reduce the gap between R_{work} and R_{free} . Side chains of residues (K88, K342, K386, K401, R691, K844, K1011) with weak electron density in the former model were deleted to decrease over-refining. The revised structural model does not alter the conformation of the complex or results of this study. The authors also revised the statistical data in the Supplementary Table 1.

-I find it rather confusing that domains have been renamed to the previous manuscripts in which Cas12i1 and Cas12i2 are structurally characterized. I highly suggest to keep the same nomenclature for things that are the same, to make comparison easier.

Authors' response:

Following the reviewer's suggestion, the authors have renamed the OBD, Helical, TSL-1 and TSL-2 domains to the WED, Helical-I, Helical-II and Nuc domains in the revised manuscript, respectively.

REVIEWERS' COMMENTS

Reviewer #1 (Remarks to the Author):

The revised manuscript from Ouyang and co-workers contains an improved Discussion section that makes a comparison of the crystal structures of Cas12i1 reported herein to previous Cas12i structures. Supplementary Figure 8c shows a superposition of the Cas12i1 pre-cleavage crystal structure reported herein to the cryoEM structures of Cas12i2. Figure 6c highlights a conformation change in the PI domain seen in this study but not observed in the previous Cas12i1 crystal structure.

A key finding is the presence of 19 bp crRNA-DNA heteroduplex in this study whereas previous Cas12i structures captured complexes with longer heteroduplexes. This seems important since it has previously been reported that the length of the crRNA-DNA heteroduplex affects the nicking versus linearizing (double strand break) activity of Cas12i (Zhang H 2020 NSMB).

Reviewer #2 (Remarks to the Author):

The authors successfully addressed all of my concerns and were able to improve the study by performing a number of additional experiments. I commend them on the exciting discoveries.

Reviewer #3 (Remarks to the Author):

Dear editor, dear authors,

The authors have addressing my main concerns. I still think the rwork/rfree gap for the Pre-cleavage R-loop complex is rather high, and myself would try to reduce this further.

I appreciate the renaming of the domains, but now the authors still refer to TSL-1 and TSL-2 in the manuscript, while it is not directly clear to me where I should see this in Figure 1 or supp figure 1 (also, there is a reference to supp figure 1g, but this panel does not appear to exist. I highly suggest to double check all references to figures to make sure the reader can find in the figure what is mentioned in the text.

If these things are considered, taken into account the major revisions made by the authors, I recommend publication of the manuscript.

Point-by-point response to Reviewers' comments

The authors thank the editor and the reviewers for accepting our manuscript as well as all valuable and helpful comments.

REVIEWERS' COMMENTS

Reviewer #1 (Remarks to the Author):

The revised manuscript from Ouyang and co-workers contains an improved Discussion section that makes a comparison of the crystal structures of Cas12i1 reported herein to previous Cas12i structures. Supplementary Figure 8c shows a superposition of the Cas12i1 pre-cleavage crystal structure reported herein to the cryoEM structures of Cas12i2. Figure 6c highlights a conformation change in the PI domain seen in this study but not observed in the previous Cas12i1 crystal structure.

A key finding is the presence of 19 bp crRNA-DNA heteroduplex in this study whereas previous Cas12i structures captured complexes with longer heteroduplexes. This seems important since it has previously been reported that the length of the crRNA-DNA heteroduplex affects the nicking versus linearizing (double strand break) activity of Cas12i (Zhang H 2020 NSMB).

Authors' response:

The authors thank the reviewer once more for all comments and suggestions, which greatly improved the quality of our manuscript.

The authors think that the 19-bp crRNA-target DNA strand heteroduplex is long enough to fully activate Cas12i1 cleaving the target DNA (TD) strand at 24-nt position after the PAM duplex.

Reviewer #2 (Remarks to the Author):

The authors successfully addressed all of my concerns and were able to improve the study by performing a number of additional experiments. I commend them on the exciting discoveries.

Authors' response:

The authors thank the reviewer for all concerns and comments, which impressed us by the professionalism.

Reviewer #3 (Remarks to the Author):

Dear editor, dear authors,

The authors have addressing my main concerns. I still think the rwork/rfree gap for the Pre-cleavage R-loop complex is rather high, and myself would try to reduce this further.

I appreciate the renaming of the domains, but now the authors still refer to TSL-1 and TSL-2 in the manuscript, while it is not directly clear to me where I should see this in Figure 1 or supp figure 1 (also, there is a reference to supp figure 1g, but this panel does not appear to exist. I highly suggest to double check all references to figures to make sure the reader can find in the figure what is mentioned in the text.

If these things are considered, taken into account the major revisions made by the authors, I recommend publication of the manuscript.

Authors' response:

The authors thank the reviewer for all suggestions, comments and careful reviewing. The revision based on your suggestions will make the manuscript easier to follow by readers.

In response to another reviewer's comments, the authors have mentioned that we have additional unpublished crystal-structure data also in the pre-cleavage R-loop state with a higher resolution (2.35Å), which has an improved $R_{\text{work}}/R_{\text{free}}$ value (0.228/0.268). The R-loop architectures in these two pre-cleavage R-loop complexes resembles with each other, and the difference between two complexes is that the 20th G-C base pair in the former complex is replaced by two unpaired 20th nucleotides A-C in the latter. The authors will soon release this unpublished crystal structure under the PDB accession code: 7EU9.

Additionally, the authors have double checked all references to figure panels. Domains previously named TSL-1 or TSL-2 for Cas12i1 are avoided in the revised manuscript. Domain named TLS for Cas12e (CasX) is kept as its name from the original article (Nature, 2019, PMID: 30718774).